# Eddy Covariance flux errors due to random and systematic timing errors during data acquisition

Gerardo Fratini[1], Simone Sabbatini[2], Kevin Ediger[1], Brad Riensche[1], George Burba[1,3], Giacomo Nicolini[2], Domenico Vitale[2], Dario Papale[2,4]

[1]LI-COR Biosciences Inc., Lincoln, 68504, Nebraska, USA
[2]Dipartimento per la Innovazione nei sistemi Biologici, Agroalimentari e Forestali - DIBAF, Viterbo, 01100, Italy
[3]R.B. Daugherty Water for Food Institute, School of Natural Resources, University of Nebraska
[4]Centro Euro-Mediterraneo sui Cambiamenti Climatici - CMCC, Lecce, 73100, Italy

*Correspondence to*: Gerardo Fratini (gerardo.fratini@licor.com)

**Abstract.** Modern Eddy Covariance (EC) systems collect high-frequency data (10-20 Hz) via instruments' digital outputs. This is an important evolution with respect to the traditional and widely used mixed analog/digital systems, as fully-digital systems help overcome the traditional limitations of transmission reliability, data quality and completeness of the datasets. However, fully-digital acquisition introduces a new problem of guaranteeing data synchronicity when the clocks of the involved devices cannot themselves be synchronized, which is often the case with instruments providing data via serial or Ethernet connectivity in a streaming mode. In this paper, we suggest that, when assembling EC systems "in-house", aspects related to timing issues need to be carefully considered to avoid significant flux biases.

By means of a simulation study, we found that, in most cases, random timing errors can safely be neglected, as they do not impact fluxes significantly. At the same time, systematic timing errors potentially arising in asynchronous systems can act effectively as filters leading to significant flux underestimations, as large as 10%, by means of attenuation of high-frequency flux contributions. We characterized the transfer function of such 'filters' as a function of the error magnitude and found cutoff frequencies as low as 1 Hz, implying that synchronization errors can dominate high-frequency attenuations in open- and enclosed-path EC systems. In most cases, such timing errors cannot be either detected or characterized a-posteriori. Therefore, it is important to test the ability of traditional and prospective EC data logging systems to assure the required synchronicity and propose a procedure to implement such a test relying on readily available equipment.

## 1 Introduction

Eddy Covariance (EC) is the most direct and defensible technique to measure atmosphere-biosphere exchange fluxes of energy and matter to date (e.g. see Aubinet et al. 2000, Aubinet et al. 2012, Baldocchi et al. 2001). The method is based on the Navier-Stokes equations for mass and momentum conservation and relies on simplifying assumptions to describe the vertical turbulent flux in terms of the covariance of the vertical wind component ($w$) and of the scalar of interest.

Calculating EC fluxes of a gaseous species requires collecting synchronous data of $w$ and of the concentration $c$ of the gas, which is typically performed, using a 3D ultrasonic anemometer and a gas analyser operating at suitable frequencies of 10 to 20 Hz. After proper data treatment and time alignment, the covariance of the two time series is calculated, from which the flux is derived (e.g. Foken et al. 2012). In this context, *synchronicity* means that $w$ and $c$ values for any given timestamp (i.e., the data that is multiplied together in the covariance) describe the properties of the same air parcel.

Regardless of the level of integration and physical configuration of the instruments within an EC system, wind and concentration data are measured by two different instruments, an anemometer relying on the speed of sound measurements

between transducer pairs, and a gas analyzer relying on the light transformations measurements in the sampling path. In addition, data collection is performed by means of a variety of more or less engineered data acquisition systems. Delays in the data flows, digital clock drifts, required separation of the measuring devices and artefacts in the data acquisition strategy can lead to poor synchronicity, i.e. to *misalignments* of the time series, such that $w$ and $c$ values assigned to a given timestamp refer to properties of fully or partially different air-parcels. If not addressed, such misalignments can lead to significant flux errors of both random and systematic nature.

Commercial solutions implementing sound engineering practices do exist for well-established EC measurements of $CO_2$ and $H_2O$ fluxes to assure a sufficient level of data synchronicity per requirements of the EC method, for selected few anemometer-analyzer pairs. However, most of such solutions are not scalable to other hardware models or gas species, because the required instrumentation doesn't necessarily support the same connectivity technology and specifications. Therefore, it is generally very challenging, for example, to simply replace a gas analyser with another one from another manufacturer and keep the same synchronization performance. Furthermore, it is customary for many research groups to assemble EC systems "in-house", especially when addressing gas species that haven't been popular enough to grow strong commercial interest. Typically, in these systems, data collection is performed with industrial data loggers or computers via serial or Ethernet connectivity, using custom-built logging software. In such cases, it is particularly important to verify that various types of data misalignment are not being introduced by the data logging system and data collection strategy to assure minimal or no bias in resulted fluxes.

In this paper, we discuss the types and sources of misalignment that can arise in poorly designed fully-digital EC systems and quantify their effects on resulting fluxes. In conjunction with site-specific characteristics, such as the typical co-spectral shapes, this information can help design appropriate data collection scheme for EC systems assembled "in-house". Users of some commercially available industrial-grade EC systems can generally assume their systems not to be affected by significant timing errors, although this can and should be verified case-by-case. We also propose a strategy for evaluating prospective EC data collection systems from the point of view of data synchronicity before they are used in routine field activities.

## 1.1 Analog vs. digital EC systems

Traditionally, a combination of analog and digital transmission systems has been used to collect EC data (Eugster and Plüss, 2010). For example, analog signals from the gas analyser were sent to an interface unit responsible for digitizing the data before merging it with anemometric data, itself coming from an A/D converter into the interface unit, typically from a sonic anemometer-thermometer (SAT). More recently, specular solutions, with the analog data from the SAT sent to an interface unit residing in the gas analyser system, became available and were widely adopted. With both of these approaches, the data was presented to the user for the flux calculation as single files with wind and gas time series merged and synchronized by the interface unit.

Analog data output allows the data to easily cross clock domains. The clock that is used to sample the original signal does not need to be synchronized to the clock that samples the analog output. This makes it very convenient to merge data from systems with unsynchronized clocks and risks of misalignments are limited to small random errors that, as we will see later, have no significant effects on fluxes.

However, collecting data in analog form has several limitations and risks. First, the number of analog channels available either as outputs from the instruments or as inputs to the interface unit is typically limited to 4 or 6, which dramatically reduces the number of variables that can be collected. In fact, historic EC raw datasets are comprised of 6 or maximum 7 variables: the 3 wind components ($u$, $v$, $w$), the sonic temperature ($T_s$) and the concentration of the gases of interest ($c$, traditionally $CO_2$ and $H_2O$); more rarely, a diagnostic variable for the anemometric data was also collected. Critical information such as the full diagnostics of both instruments and their status (e.g. the temperature and pressure in the gas analyser cell or the signal strength) or the original raw measurement (speed of sound, raw data counts etc.) are not collected in most analog systems, limiting the

means for quality screening and limiting the possibility of future re-computation of the most fundamental raw measurements. Another problem with analog data collection is that signals are subject to degradation due to dissipation, electromagnetic noise and ageing of cables and connectors, which reduces the quality of collected data (Barnes, 1987). In addition, although all raw measurements are analog in nature, they are typically immediately digitized (*native* digital format provided by the manufacturer) and then - in case of analog data collection - they are re-converted to analog, sent to the interface unit and there converted back to digital; these A/D-D/A conversions potentially degrade the signal adding noise and dampening high-frequency signal components (Eugster and Plüss, 2010). For these reasons, analog connectivity should nowadays be avoided whenever possible in favor of fully-digital solutions.

In fully-digital EC acquisition systems both data streams are collected in their respective *native* digital format, i.e. without additional A/D conversions other than those implemented by the manufacturer to provide digital outputs. Fully digital systems largely or completely overcome both problems with mixed analog/digital systems, using more robust and less corruptible data transmission protocols, and providing the possibility of collecting all variables available from the individual instrument. However, combining digital data streams from different instruments brings new challenges, most notably with respect to data synchronization. While moving between clock domains is trivial in an analog system, it can be much more challenging with digital data when the involved clocks can be completely asynchronous to each other.

### 1. 2 The problem of clock synchronization in digital systems

Different strategies exist for collecting data digitally. First, instruments can perform the measurements according to their own scheduler or a trigger. In the scheduler case, data can then be collected by polling the instrument for the latest available data (polling mode) or by keeping an open channel where data is streamed to (streaming mode). In the trigger case, data is usually made available after a fixed or somewhat variable delay to the logging device. This delay is due to the acquisition time and could also include a delay due to filtering. However, the timestamp for the acquired data is assigned based on the occurrence of the trigger, therefore removing any timing error due to that delay. All modes have advantages and disadvantages. As described later, triggering and polling modes are less susceptible to timing errors, but they require the instrumentation to be designed for the particular triggering or polling system adopted. They are therefore best suited for EC systems built with components all from the same manufacturer. For the same reason, such systems are commonly not flexible enough to accommodate 3rd-party instrumentation. EC systems commercialized by Campbell Scientific Inc. (Logan, UT, USA, "CSI" hereafter) are examples of integrated systems using a data triggering strategy to collect data from instruments designed ad-hoc. Data communication in these systems is realized via the SDM protocol (or its evolutions), which is a CSI proprietary protocol, implemented only in CSI instruments and some $CO_2/H_2O$ gas analysers by LI-COR Biosciences Inc. (Lincoln, NE, USA, "LI-COR" hereafter). By contrast, most instrumentation available for fast wind and gas measurement provides data transmission options only in streaming mode. As a consequence, most data logging solutions developed by the scientific community or by commercial entities are designed to handle data provided in streaming mode and are therefore flexible to accommodate a wide variety of instrumentation. Examples of such logging systems developed by the community are PC-based software such as Huskerflux (https://github.com/Flux-Dave/HuskerFlux), EddyMeas (Kolle and Rebmann, 2007), EdiSol (EdiSol User Guide V0.39b https://epic.awi.de/29686/1/Mon2005d.pdf), or the already referenced system proposed Eugster and Plüss (2010) for EC measurement of methane. As for commercial solutions, LI-COR provides industrial-grade EC systems based on the SmartFlux® system, that can accommodate a wide variety of instrumentation using the data-streaming approach. However, collecting data in streaming mode exposes the risk of introducing significant timing errors, because of the number of asynchronous *digital clocks* involved.

Digital clocks are electronic oscillator circuits that use the mechanical resonance of a vibrating crystal of piezoelectric material to create an electrical signal with a precise frequency, which is then used to keep track of time. The number and quality of clocks involved in an EC system vary with the data collection strategy and technology. In systems based on data triggering or polling, there is only one critical clock (the one responsible for the timing of the triggering or polling signal, respectively), therefore there is no significant risk of introducing systematic timing misalignments between data from different instruments (see later). With these systems, the risk is limited to random and/or constant misalignments. As we will see later, random or constant misalignments do not entail large errors. For this reason, in the remainder of this section we consider more in details the situation with systems based on data streaming.

In such systems the clocks potentially relevant are, in general:

- The sampling clocks of the sensing instruments (in the typical EC system, the SAT and the gas analyser), responsible for sampling data at the prescribed rate with sufficient precision and accuracy;
- For systems that transmit data serially (RS-232 or RS-485), the serial clock of the same sensing instrument, which may or may not be *correlated* to its sampling clock;
- The clocks of the logging device(s) (data logger, PC, etc.), responsible for attaching a timestamp to the data. If a single logging device is used this is usually also responsible for merging data streams from the different instruments; in case dedicated logging systems are used for different instruments, merging is performed in post-processing and the clocks of the different loggers must, therefore, be aligned sufficiently frequently (e.g. every second, using a GPS signal).

Typical open digital communication protocols used for EC instruments with data-streaming instrumentation are serial (RS-232, RS-485) and packet-based data protocols (Ethernet). In devices that transmit data via serial communication, such as SATs, there are no means to synchronize the sampling clock of the device to that of the data logger. With such devices, the best that can be done is to assign a timestamp *after* transmission, based on the clock of the data logger (this last step should be performed carefully to avoid large inaccuracies due to serial port latencies, especially in PC-based systems). In addition, devices implementing serial communication have an asynchronous clock that drives those protocols (e.g. Dobkin et al., 2010). If this clock is *correlated* with the device's sampling clock, the receiving data logger can - at least in principle - reconstruct the sampling clock. However, in devices that do not correlate sampling and serial clocks (such as those that output data in a software thread that is independent of an acquisition thread), the system scheduler then determines when data is transmitted, thereby completely isolating the serial clock from the sampling clock and making it impossible for the data logger to reconstruct the sampling clock.

Packet-based data communications such as Ethernet even further isolate the sampling clock from the transmission clock. In devices using this protocol, it is therefore impossible to reconstruct a sampling clock. However, for Ethernet-based systems additional protocols are available, such as NTP or PTP, to actually synchronize all system clocks. The synchronized system clocks then allow the data to be correctly timestamped *before* transmission, eliminating any synchronization issue, provided that downstream software can align the various data streams based on their timestamps (e.g. Mahmood et al., 2014).

**1.3 Types of timing errors**

In typical EC data acquisition setups, time series collected by different instruments can show three distinct types of misalignments (Figure 1):

*Time lags:* these are *constant* offsets in otherwise perfectly aligned time series. They can be the result of constant electronic delays or of fixed delays due to digital signal processing (DSP). More frequently, they result from a physical separation of the sampling volumes or from the delay due to the time needed for the passage of air in a sampling line.

***Random timing errors*** (RTEs in the following) occur when the timestamps assigned to the data differ from the exact time dictated by the nominal sampling frequency, and such differences are *randomly distributed* so that, on average, the actual frequency is equal to the nominal one. In practice, in the EC context, it is more useful to consider the random differences in the timestamps assigned to data from one instrument with respect to that of the paired instrument. In fact, in the hypothetical case in which the two instruments would have the exact same sequence of random errors, that would not introduce any misalignment and hence no flux bias.

***Systematic timing errors*** (STEs) occur when the timestamps assigned to the data differ from the exact time dictated by the nominal sampling frequency, and such differences are *systematic*, e.g. the actual time step is slightly longer or shorter than the nominal one for time spans of the order of the flux averaging interval. Again, in EC we are only concerned with systematic *relative* errors, for identical errors in the two concerned instruments would entail no misalignment and hence no flux bias. Instances of each type of misalignment can, and typically will, be present at the same time to various degrees.

## 1.4 Sources of misalignment and their effects on time series

### 1.4.1 Spatial separation between sampling volumes

In a SAT, the *sampling volume* is the volume of air between the upper and lower sets of transducers. Similarly, in an open-path gas analyser such as the LI-7500 $CO_2/H_2O$ analyser and the LI-7700 $CH_4$ analyser (LI-COR Biosciences Inc., Lincoln, NE, USA), the sampling volume is the volume of air between the upper and lower mirrors. In a closed- or enclosed-path gas analyser such as the LI-7000, the LI-7200 (LI-COR Biosciences Inc.) and the EC155 (Campbell Scientific Inc., Logan, HT, USA), instead, the sampling volume can be identified with the volume of the intake device, e.g. a rain cup.

Even in the hypothetical situation of perfectly synchronized timestamps for wind and gas data, if the respective instruments' sampling volumes have to be spatially separated to avoid presently intractable flow distortion issues in the anemometer, as is notably the case with open-path setups (see, for example, Wyngaard, 1988; Frank et al., 2016; Horst et al., 2016; Grare et al., 2016 and Huq et al. 2017), the corresponding time series will be affected by misalignment, possibly to varying degrees. Indeed, assuming the validity of Taylor's hypothesis of frozen turbulence, wind and concentration data will be affected by a time-lag (the time air takes to travel between the two sampling volumes), which will be further modulated by wind intensity and direction. Additionally, modification of turbulence structure intervening while air parcels transit through the dislocated instrument volumes may introduce further uncertainty in flux estimates (Cheng et al., 2017). In case of co-located sensors (e.g. Hydra-IV, CEH; IRGASON, Campbell Scientific Inc.) this problem is not present but is replaced by the flow distortion issues mentioned above and not addressed in the present study.

### 1.4.2 Spatial separation between measuring volumes

In a SAT, the sampling volume coincides with the measuring volume, i.e. wind velocity is measured exactly where it is sampled. The same is true for open-path gas analysers. However, closed-path and enclosed-path analysers take the sampled air into a measuring cell via a sampling line that can be anywhere between 0.5 to 50 meters long, with its inlet usually placed very close to the SAT sampling volume. This implies a delay of the time series of gas concentrations with respect to the wind time series. Such delay can be more or less constant in time depending on the possibility of actively controlling the sampling line flow rate. In systems without flow controllers, the flowrate may vary significantly in response to power fluctuations or tube clogging and so would the corresponding time-lags.

### 1.4.3 Clock errors

Quartz crystal clocks universally used in electronic devices are subject to two main types of error: periodic jitter and frequency drift (McParland, 2017).

*Period jitter*

Period jitter in clock signals is the random error of the clock with respect to its nominal frequency. It is typically caused by thermal noise, power supply variations, loading conditions, device noise, and interference coupled from nearby circuits. Jitter is a source of RTE in time series.

*Frequency drift*

The oscillation frequency of a clock varies with temperature, leading to *drifts* of the measured time and hence to STEs. The
drift of a clock can be expressed as the amount of time gained (or lost) as a result of the drift per unit of time, with suitable units being µs/s. For example, a drift of -30 µs/s means that a clock accumulates 30 µs of delay per second, or about 2.6 seconds over the course of one day ($2.6 = 30 \cdot 10^{-6} \cdot (24 \cdot 60 \cdot 60)$). The dependence of crystal oscillation frequency on temperature varies, even dramatically, with the type and angle of crystal cut and can be modeled as quadratic (BT, CT, DT cuts) or cubic (AT cuts) (Hewlett Packard 1997). Figure 2 shows exemplary drift curves for different crystal cuts. Typically, the nominal
frequency (e.g. 32 kHz) is specified at 20 or 25 °C. Apart from that temperature, the frequency can vary for example according to (for a BT cut):

$$\frac{f - f_0}{f_0} = -\alpha (T - T_0)^2 \cdot 10^{-6} \tag{1}$$

where $f_0 = f(T_0 = 25\ °C)$ and typical values of $\alpha$ range 0.035-0.040.

Clocks in EC systems can be exposed to large variations of temperature (day-night, seasonal cycles). Because we are concerned with *relative* drifts, we are interested in *differences* in the temperatures experienced by the instruments' sampling/logging clock as well as with differences in their temperature responses. Clocks experiencing similar temperatures and with similar temperature responses, would minimize relative drift. On the contrary, clocks with opposite responses to temperature will
result in relative drifts that are close to the sum of the individual drifts, e.g. in the case of AT-cut crystals with different angles of rotation at relatively high temperatures (i.e. above 30 degC, see Figure 2)

It is also to be noted that temperature-compensated clocks do exist, which have accuracies of around $\pm 2$ µs/s. As we will show later, such drifts can be safely neglected, as long as clocks are synced sufficiently often (e.g. once a day). For completeness, we note that clock drifts also occur due to the ageing of components. However, the absolute values of typical ageing rates (<
1 µs/year) are of no concern in EC applications. Because STEs in EC systems are caused primarily or exclusively by clock drifts, in the rest of the paper we will use the terms STE and *drift* interchangeably.

### 1.4.4 Further sources of timing errors in digital asynchronous systems

*Connectivity*

Ethernet connectivity available in commercial loggers and industrial PCs (e.g. SmartFlux 2 and 3 by LI-COR Biosciences Inc.,
CR3000 and CR6 by Campbell Scientific Inc.) can be used for data acquisition in EC systems. The acquisition is usually done using the Transmission Control Protocol (TCP), a packet-based protocol specifically designed to preserve the accuracy of the data during transmission. TCP is however not designed to preserve the temporal aspect of the packets. The TCP receiving system must buffer up packets and signal the sender if an error occurs in any packet. This can cause packets to even arrive out

of order, even though they are always delivered to the application in order. Therefore, TCP-based systems are subject to significant RTEs and, potentially, to STEs.

Serial communication devices typically use a first-in-first-out (FIFO) policy to buffer data, on both the sending and the receiving sides. The FIFO increases the efficiency and throughput by reducing the number of interrupts the CPU has to handle (e.g. Park et al., 2003). Without a FIFO buffer, the CPU has to interrupt for every data unit. With a FIFO, the CPU is interrupted only when a FIFO is full, or a programmed amount of data is ready. However, the FIFO can become a problem on a system where it's desired to correlate the serial clock to the sampling clock. If not properly handled, the FIFO introduces timing jitter on both the transmitter and the receiver, hence inducing RTEs in the system.

*Time response*

In a streaming-based system, an instrument with a time response (irrespective of the supported output rate) slower than the sampling rate will lead to RTEs even in the absence of any clock errors, because the measurement cannot in general be performed at the required moment in time. In the general case, such an instrument will be oversampled (i.e. the same measured value will appear multiple times in the final time series). Eugster and Plüss (2010) discuss in detail the consequences of such occurrence with a $CH_4$ gas analyser with a time response of 5.7 Hz in a system sampling data at 20 Hz, concluding that the flux errors are negligible in most applications.

For completeness, we note that in virtually all EC instruments the native measurement is time-discrete. For example, in non-dispersive infrared (NDIR) gas analyser, the presence of a rotating filter used to multiplex the desired infrared bands makes the gas concentration measurement frequency dependent on the wheel rotational frequency, which leads to RTEs. Nonetheless, if the rotational speed is high enough (e.g. > 100 Hz) the resulting errors are minimal.

**1.5 Dealing with timing errors in EC practice**

The fundamental difference between time-lags on one side and RTE/STE on the other side is that constant time-lags can, at least in principle, be addressed *a posteriori* during data processing. The topic of correctly estimating and compensating time-lags has long been discussed in the EC literature (Vickers & Mahrt 1997; Ibrom et al., 2007; Massman and Ibrom, 2008; Langford et al, 2015), and corresponding algorithms are available in EC processing software. We will therefore not further discuss time-lags in this paper.

Random and systematic timing errors, instead, are not identifiable and therefore it is not possible to correct them. However, their effect on flux estimates, as we will see, can become significant. For this reason, the only viable strategy to reduce flux biases is to design the data acquisition system in a way that prevents or minimizes the possibility of their occurrence.

The focus of this paper is, therefore, the quantification of flux underestimations as a function of RTE and STE, so as to derive quantitative specifications for a data acquisition system that minimizes EC flux losses. We further propose a simple scheme for evaluating existing data acquisition systems with respect to data synchronization.

**2 Materials and Methods**

**2.1 Simulation design**

In order to accurately quantify how time-alignment errors affect flux estimates, we performed a simulation study. As a reference, we used the covariance estimated from high-frequency data of vertical wind speed ($w$) and sonic temperature ($T_s$) which are by definition perfectly synchronized since they are computed starting from the same raw data (the travelling time of sound signals between pairs of transducers in a SAT). We also assumed that high-frequency time series are provided at perfectly constant time steps of 0.1 (10 Hz) or 0.05 (20 Hz) seconds. Subsequently, we manipulated the array of timestamps at which the sonic temperature data was sampled in order to simulate realistic ranges of RTEs and STEs. Values of sonic

temperature in correspondence to the new, simulated timestamps were estimated by linearly interpolating the closest data points in the original series (Figure 3).

Before calculating covariances, standard EC processing steps were applied such as spike removal (Vickers and Mahrt, 1997), tilt correction by double rotation method (Wilczak et al., 2001) and fluctuations estimation via block-averaging. Covariance estimates obtained with the new versions of $T_s$ were then compared with the reference to quantify the effect of simulated timing errors. Flux biases would be almost identical to biases in covariances, weakly modulated by corrections intervening between covariance and flux computation, such as spectral corrections and consideration of WPL effects (e.g. Fratini et al. 2012). The simulation study was implemented in the source code of EddyPro v6.2.1 (LI-COR Biosciences Inc, Lincoln, NE; Fratini and Mauder, 2014).

For the present analysis, we simulated RTEs ranging ±1 to ±100 ms, that is, up to the same order of the sampling interval. As an example, with a simulated ±10 ms RTE using 10 Hz data (nominal time step = 100 ms), simulated time steps for Ts varied randomly between 90 and 110 ms, with an average of 100 ms. We note that RTEs of 10-100 ms won't usually be caused by clock jitter, which is typically several orders of magnitude smaller but can easily be caused by acquisition systems based on serial or Ethernet communication not specifically designed to collect synchronous time series, as described above.

For STEs, we simulated relative drifts ranging 10 to 180 µs/s (specifically 10, 30, 60, 90, 120, 150 and 180 µs/s). For example, to simulate a STE of 60 µs/s we kept the original $w$ time series (time step equal to 100 ms) and modified the time step of $T_s$ to be 100.006 ms. This may seem a negligible difference, which however accumulates a difference of 108 ms between $w$ and $T_s$ within 30 minutes and manifests itself as a difference of one row in the length of the time series (i.e. 18000 value for w and 17999 for Ts @ 10 Hz). Similarly, systematic errors of 120 and 180 µs/s would lead to 2 and 3 row differences, respectively.

Systematic timing errors such as frequency drifts essentially act as low-pass filters, which can be described by characterizing their transfer function, provided that the drift is known, as in our simulation. Here, for each 30-min period and for each STE amount, we calculated an *in-situ* transfer function as the frequency-wise ratio of drifted and original $w$-$T_s$ cospectra:

$$TF\ (f_n) = \frac{co(w, T^{STE} \mid f_n)}{co(w, T \mid f_n)}, for\ STE\ =\ [10, \ldots, 180]\ \mu s/s \tag{2}$$

where $T^{STE}$(K) is the simulated sonic temperature for each STE value, $f$ (Hz) is the natural frequency and $TF\ (STE \mid f_n)$ is the *in-situ* transfer function for STE. We repeated this calculation for a number of cospectra ranging 1000-2000 (depending on data availability). The ensemble of all transfer functions so obtained and for each STE amount was then fitted with the following function, that was found to reasonably approximate the data obtained for all drifts at all sites in the most relevant frequency range ($0.01 - 5$ Hz):

$$TF(f \mid f_n) = (1 + \beta)\ \frac{1}{1 + \left(\frac{f}{f}\right)^\alpha} - \beta \tag{3}$$

where:

$f$ (Hz) is the transfer function cutoff frequency and $\alpha$ and $\beta$ are fitting parameters whose values were found to vary very little around $\alpha = 2.65$ and $\beta = 0.25$.

## 2.2 Datasets

We performed simulations on 4 datasets acquired from EC sites representative of various ecosystem types and climatic regimes and characterized by the different height of measurement and height of the canopy:

- *IT-Ro2*: a deciduous forest of Turkey Oak (*Quercus cerris* L.) in Italy. Eddy covariance measurements were carried out from 2002 to 2013 and the period used for the simulations was May 2013, when the canopy height was 15 m and the measurement height 18 m (Rey et al., 2002).

- *IT-Ro4*: located at about 1 km from IT-Ro2, this site is a rotation crop where EC measurements have been carried out from 2008 to 2014. Data used for the simulations include 43 days in 2012 when crimson clover (*Trifolium incarnatum* L.) was cultivated (maximum canopy height 60 cm, measurement height of 3.7 m)

- *DK-Sor*: evergreen forest near Sorø, DK. EC measurements are performed since 1997: during the period used for the simulation (the entire 2015) the forest was 25 m tall and the EC system placed at 60 m a.g.l. (Pilegaard et al., 2011).

- *IT-CA3*: fast-growing, short-rotation coppice (SRC) of poplar clones planted in 2010, located in Castel d'Asso, Viterbo, Italy. The EC tower was installed at the end of 2011, and measurements were taken until mid-2015. The period used for the simulation included 9 months over the period 2012-2015, with a canopy height ranging 0-5.3 m, and the measurement height between 3 and 5.5 m (Sabbatini et al., 2016).

## 2.3 Validation of the simulation design

Although the proposed simulation design enables the evaluation of resulting errors using readily available EC data, we note that interpolating data sampled at 10 or 20 Hz frequency can potentially introduce artefacts (due to the lack of information at higher frequencies) such as, for example, an undue reduction of the sonic temperature variance, which would result in artificial reduction of the $w$-$T_s$ covariance. In order to detect any such effects, we preliminarily implemented a validation procedure, making use of one week of sonic data from a Gill HS (Gill Instruments, Lymington, UK) collected at 100 Hz. The validation involved the following steps:

1. Subsampling at 10 Hz and simulating timing errors as described above, i.e. interpolating starting from the subsampled data.
2. Subsampling at 10 Hz and simulating timing errors by interpolating the original 100 Hz data.
3. Comparing $w$-$T_s$ covariances obtained in 1 and 2.

The timing errors simulated interpolating the original 100 Hz measurements (option 2 above) are much less prone to artefacts because interpolation occurs between data that are 0.01 seconds apart, an interval too short for any significant flux signal to occur. Using this procedure, we could verify that there is no detectable difference between results obtained with 100 Hz and 10 Hz data (not shown), which implies that the interpolation procedure is not introducing significant artefacts in the estimation of variances and covariances and therefore the simulation can be performed with virtually any historic EC dataset using the available code.

## 3 Results and discussion

Figure 4 compares covariances $w$-$T_s$ obtained with increasing amounts of RTE against the reference covariance obtained with the original, perfectly synced, time series. Reduction in covariance estimates is fairly negligible provided that RTE is of the same order of magnitude of the sampling interval or less. Largest discrepancies were observed for the IT-CA3 and IT-Ro4 sites with a covariance underestimation of 3% for RTE of amplitude 100 ms. As mentioned earlier, such large timing errors are never the result of electronic clock jitter and may instead be caused by a data transmission system not designed for time synchronization, such as TCP.

Conversely, flux biases induced by systematic timing errors are both more significant and more variable. Figure 5 shows that a STE of 60 µs/s (1 row of difference in a 30-minute file with data collected at 10 Hz) can lead to errors anywhere between 0 – 4%, increasing to 1 – 8% for a STE of 120 µs/s (2 rows of difference) and to 1 – 11% for a STE of 180 µs/s (3 rows of difference).

Figure 6A shows an example of the transfer functions derived using the procedure described in Sect 2.1. The Figure refers to the site IT-CA3, but the filters obtained for the other sites had very similar characteristics, as illustrated in Fig. 6B using the mean cutoff frequencies computed for all sites at each STE amount: the tight ±3.5σ range merely demonstrates that the low-pass filter properties of the STE are independent from the data used to derive it, and only vary with the error amount.

Nonetheless, in Figure 5 we showed how the same STE leads to very different flux underestimations at a different site. For example, a systematic error of 180 µs/s led to flux biases of 1% and 11% at IT-Ro2 and IT-Ro4, respectively.

The reason is related to the distribution of the flux contributions across the frequency domain. The more the flux cospectrum is shifted towards higher frequencies, the more it will be dampened by any given low-pass filter and the higher the resulting flux bias will be. In other words, systematic timing errors are a source of high-frequency spectral losses, not dissimilar to the ones traditionally considered in EC (Moncrieff et al. 1997; Massmann, 2000; Ibrom et al. 2007). Figure 7 depicts the low-pass filtering effects of several STEs as applied to three different hypothetical EC systems, characterized by different "initial" cutoff frequencies (caused by other sources of attenuation such as e.g. length of the sampling line) deployed in two contrasting scenarios (high vs low measurement height). It is evidenced that at high measurement heights effects are negligible irrespective of the "original" cutoff frequency of the system (a-c). The reason is that the STE filters act on cospectra that are shifted to low-frequencies and have therefore very low high-frequency content. At low measurement height, instead, STEs significantly increase spectral losses if the system has a high initial cutoff frequency (e-f), while if the system as a poor initial spectral response (d), STEs are irrelevant because, again, high-frequency co-spectral content is minimal to start with.

To put this new source of high-frequency losses in perspective quantitatively, we note that for EC systems based on an enclosed-path gas analyser (LI-7200), cutoff frequencies ranging 1.1 Hz (for less optimized) up to 7-8 Hz (for systems with optimized intake rain cup and heated sampling line) were reported in literature (e.g. Fratini et al. 2012; De Ligne et la., 2014; Metzger et al. 2016). Similar values are usually found in systems based on open-path setups. Significant STEs can thus easily become leading sources of flux biases in modern EC systems deployed at low measurement heights and/or very limited spectral losses due to other causes, with the additional complication that they are hard to detect and quantify. In fact, once acquired and stored in files, it is generally not possible to establish whether a drift between data streams occurred. Missing lines in one data stream could be either filled in by the data acquisition software (e.g. by means of the "last observation carried forward" technique) or could be compensated by dropping one line in the paired, longer series. In both cases, one would be unable to detect the problem, which is however obviously not solved by these solutions, meant only to build a complete rectangular dataset. On the contrary, a mismatch of one or two lines in the length of the time series is *not* necessarily the sign of an occurring STE, as it could also be the result of an imperfect timing in opening/closing a data stream, or some combination of the two factors. For these reasons, it is very difficult, if not practically impossible, to detect STEs, distinguish them from other timing errors or artefacts and, more importantly, to infer the type and amount of error that is being introduced in the covariances. The only sign of a potential timing problem is an attenuated cospectrum, as evaluated with respect to an available reference or model. But from the cospectra attenuation alone, it is impossible to establish the presence of a timing error and, even more, disentangle it from other sources of attenuation. The only possibility is thus to estimate an ensemble spectral correction based on cospectra, which would correct only sources of errors without the ability to discriminate them, which is less than ideal (e.g. Ibrom et al., 2007). It is therefore advised to evaluate the performance of a data acquisition system before it is put in operation.

*Evaluating synchronicity of an EC data acquisition system*

If both EC instruments can receive analog inputs, a possible way to evaluate synchronicity in the data logging system is to connect a signal generator to both EC instruments (SAT and gas analyser) and collect the data via the data logging system in the configuration that would be adopted in normal operation (Figure 8). The result is 2 replicates of the known signal data,

whose timestamps will in general not be synchronized, in the sense that the same nominal timestamp will be attached to two different pieces of data. The two datasets can then be compared to calculate the phase difference between the clocks and hence assess RTEs and STEs. For example, a cross-correlation of the time series yields the phase difference. It's important to test the time series at different time intervals, e.g. ½ hour for several days, in actual field conditions that undergo significant temperature variations. The two instruments are synchronized if the cross-correlation yields the same result every time. If this constant phase offset (time-lag) is different from zero, this measurement quantifies signal delays in a system, which can be addressed either by optimizing the data logging system or by taking this offset into account while setting up the time-lag automatic computation in post-processing. A cross-correlation that changes over time, instead, is a strong indication of occurring STEs. It is very difficult to anticipate the evolution in time of the phase change as it depends on the clock's crystal cuts, quality and temperature sensitivity. In general, we may expect a linear trend in the phase if temperatures don't vary strongly (see also later) or a trend modulated by a diurnal pattern if temperature plays a role.

A simpler (though less controlled) option, in case a signal generator is not accessible, is to use the analog outputs from at least one of the EC instruments. In this case the test involves transmitting (at least) one of the analog outputs to the analog input of the companion instrument (e.g. $w$ sent via analog output of the SAT to an analog input channel of the gas analyser). In this way the raw-data files contain 2 replicates of that variable, each collected according to the timing of the respective instrument: the timestamps of the digital version is logged according to timing of the sending instrument (SAT, in the example), while those of its analog version are logged according to the timing of the receiver (gas analyser). The same cross-correlation analyses described above can then be performed.

In both versions of the tests, results may be affected by minor RTEs if the various A/D or D/A tasks are not accurately synchronized with the measurement and serial output tasks. However, such RTEs should not affect the ability to detect and quantify occurring STEs.

Note also that, in both versions of the test, the way raw data is stored may have a strong impact on how to interpret the results. For instance, depending on the specifics of the data acquisition system, collecting a unique file with 3 days' worth of data or collecting 30-min files for three days, can provide different results, e.g. because the act of closing a file and opening a new one can cause the data streams to be partially or completely "realigned".

To exemplify the test, we collected about 3 days of 20Hz wind data from a SAT (HS-100, Gill Instruments Ltd., Lymington, UK) both in native digital format (via a RS-232 port, indicated with the subscript $d$ in the following) and in analog format via the A/D of a LI-7550 Analyser Interface Unit (LI-COR Biosciences Inc.) which was then collected via a second RS-232 port (indicated with subscript $a$), using an industrial-grade PC running Windows XP. Thus, the data logging system under testing was "a Windows PC collecting EC instruments via RS-232, which were setup to transmit data in streaming mode". The two data streams were completely independent to each other, and we attached timestamps to the records based on the Operating System clock as the data was made available from the serial port to the application collecting the data. We then merged the two datasets based on timestamps and split the resulting 3-day file into 30 minutes. Finally, we calculated time-lags between pairs of homologous variables (e.g. $u_d$ vs $u_a$, but results were identical for all anemometric variables), which are shown in Figure 9. The linearity of the data suggests that the system is affected by a fairly constant STE of about 50 μs/s, as quantified by the slope of the line.

Using the same setup, we further collected 2 days of data directly stored as 30-min files and again computed time-lags between homologous variables. The acquisition system was able to realign the two series at the beginning of each half hour resetting the time-lag between them to roughly zero. Nevertheless, calculating time-lags on overlapping 5 min periods, we found that *within* each half-hourly period the time-lags increased of 0.05 s (32% of the times) and of 0.1 s (68% of the times), which again indicates an STE ranging 30-60 μs/s which, as shown above (Fig. 5), can lead to detectable flux biases. We stress that

the system used in this experiment was not optimized for data acquisition and the aim was solely that of evaluating the proposed test.

## Conclusions

Undoubtedly, modern EC systems should log high-frequency data in a native digital format, so as to collect all possible measurement, diagnostic and status information from each instrument and assure the creation of robust, self-documented datasets, which are essential to the long-term research goals of climate and greenhouse gases science.

Commercial data acquisition solutions exist, that one can legitimately expect to ensure a proper data synchronization, such as the SmartFlux® system by LI-COR or the SDM-based system by CSI. There exist also applications developed by research institutions that specifically address the synchronization issue. In all these cases it is however possible to test the synchronization in order to confirm the expected performances.

When dealing with novel gas species, however, assembling EC systems from instrumentation that is not necessarily designed to be integrated is often the only choice available to the researcher and "in-house" solutions become necessary. In such cases, extreme care and expertise must be put in the handling of different digital data formats and transmission mode, in a context where data synchronicity is essential. We have shown that failure to do so can result in significant biases for the resulting fluxes, which depend on the type of timing error (random or systematic) and its amplitude, as well as on the co-spectral characteristics at the site. We have also explained how such errors are virtually impossible to detect and quantify in historic time series. It is, therefore, necessary to avoid them upfront, via proper design and evaluation of the data logging system.

Deploying a simple testing setup that makes use of equipment usually available to the EC experimentalists, we demonstrated how, for example, a naïve data collection performed asynchronously on a Windows XP industrial-PC leads to significant relative drifts among the time series, which is bound to generate flux underestimations. With minor *ad hoc* adjustments, the same testing setup can be used to evaluate any EC data logging system. While evaluation of existing systems was beyond the scope of our work and we do expect synchronization issues to be more of a risk for "in-house" solutions, the proposed testing setup for evaluating data synchronization applies equally to "in-house" and to commercial solutions and we do invite researchers and companies to test their systems.

With this in mind, we recommend the scientific community to promote collaboration and synergy among manufacturers of EC equipment, for technological solutions that *guarantee* sufficient synchronicity do exist - such as Ethernet connectivity deploying the PTP protocol - but, in order to be utilized, they require all instrumentation to be compatible with those technologies, which is yet not still the case.

A final note on the data collected until now and largely shared and used in publications. As stated above it is impossible to detect the presence of a synchronization issue on archived dataset. However fully-digital acquisition in streaming mode started to be largely adopted only recently and this limits the potential impact of the issue on historical data. In addition, as also explained in the results, the effect of a STE acts as a spectral loss and hence it may be (at least partially) compensated for and corrected by in-situ spectral corrections based on cospectra.

## Acknowledgments

DP, GN and DV thank the ENVRIplus project funded by the European Union's Horizon 2020 Research and Innovation Programme under grant agreement 654182 and the RINGO project funded under the same program under grant agreement 730944. SS thanks the COOP+ project funded by the European Union's Horizon 2020 research and innovation programme under grant agreement No 654131.

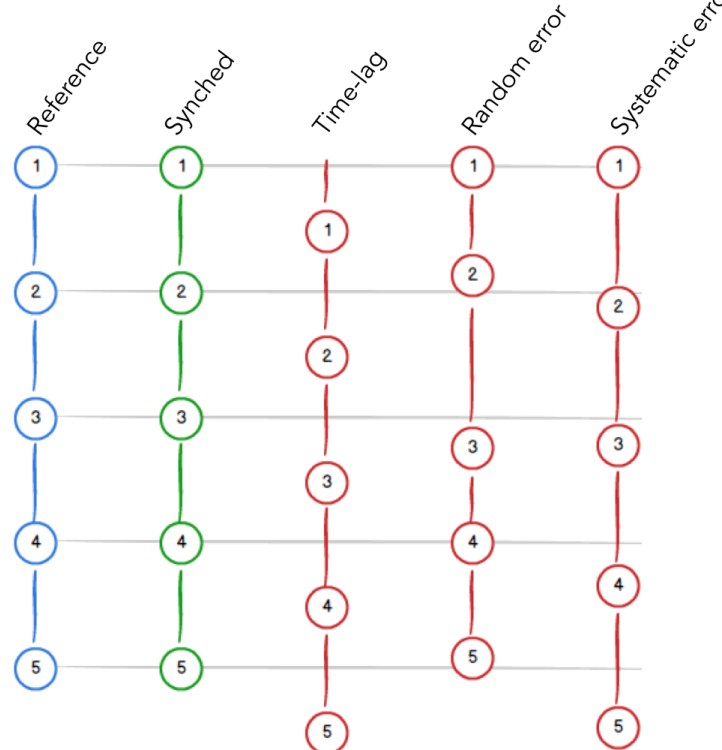

**Figure 1: Schematic of the three types of misalignments common in EC data. Given a reference time series (e.g. *w*, blue), the ideal paired gas concentration time series is perfectly aligned (green). The three red time series exemplify (from left to the right): a constant offset (time-lag), random variations around the perfect alignment (random error) and a systematically larger time step (systematic error). Real data is typically affected by a mix of all error types in varying amounts.**

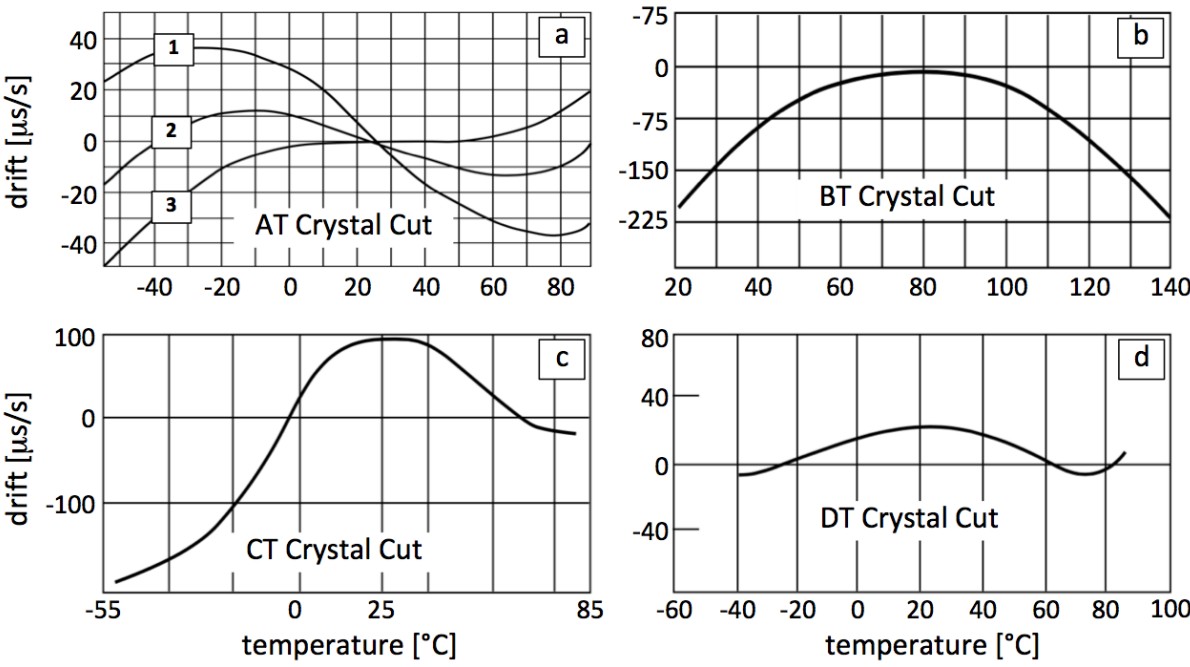

**Figure 2: Exemplary temperature dependence curves for clocks with AT (a), BT (b), CT (c) and DT (d) cuts. In (a) the 3 curves refer to different angles of rotations of the crystal. Reproduced and adjusted from Hewlett Packard (1997).**

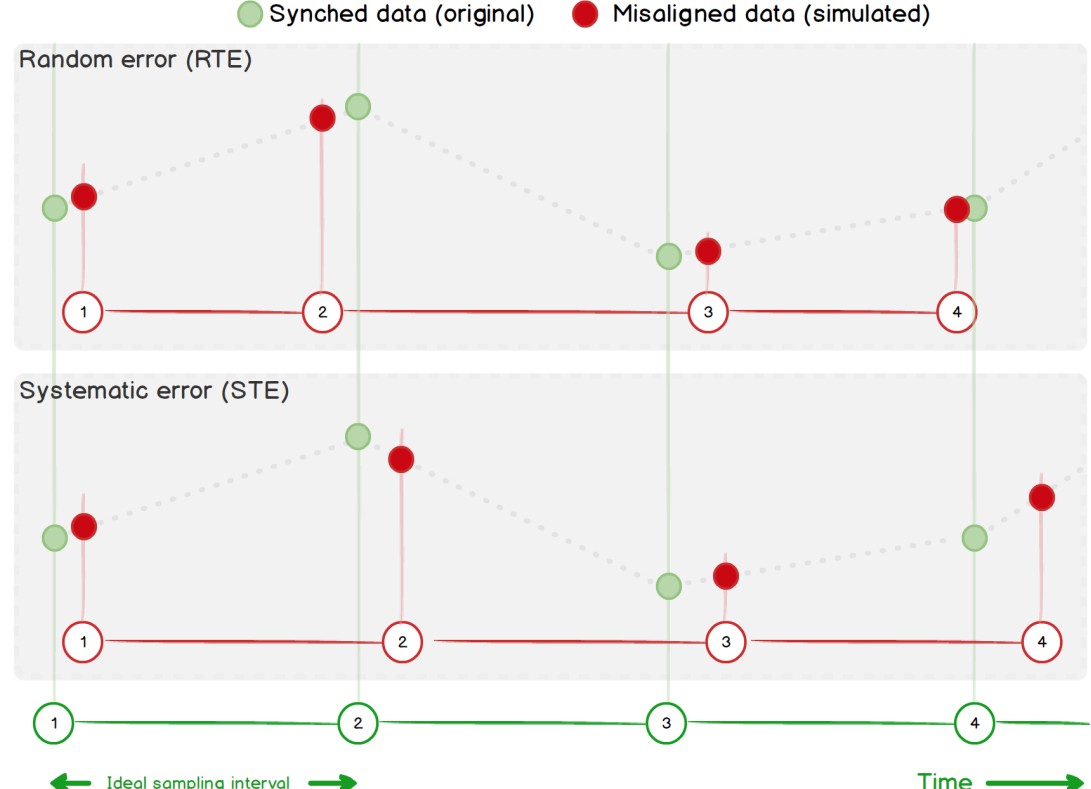

**Figure 3: Sketch of the timing error simulation via linear interpolation. Green dots represent original $T_s$ data points, equally spaced at the prescribed (nominal) time steps. In the upper panel, RTEs are simulated as time steps randomly varying around the nominal value. In the lower panel, STEs are simulated as a time step larger than the nominal one, whereby the difference between the assigned and correct timestamps always increase in time.**

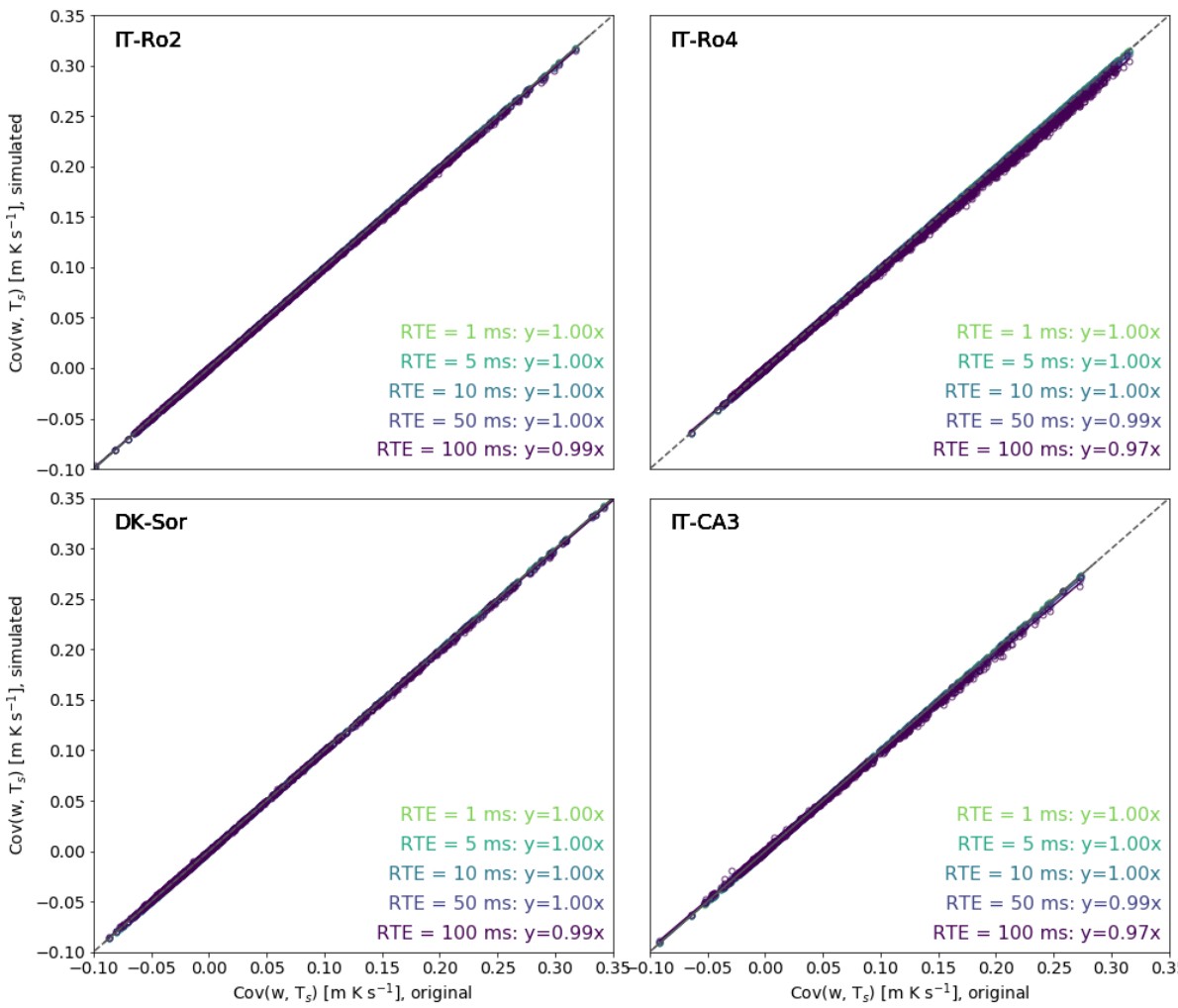

**Figure 4: Simulation of RTEs.** Covariances *w-T$_s$* obtained with a set of simulated random errors of different amplitudes (y-axes) are compared to the covariances *w-T$_s$* computed with the original time series (x-axes), for the four sites. All regressions had offset equal to zero and r$^2$ > 0.99.

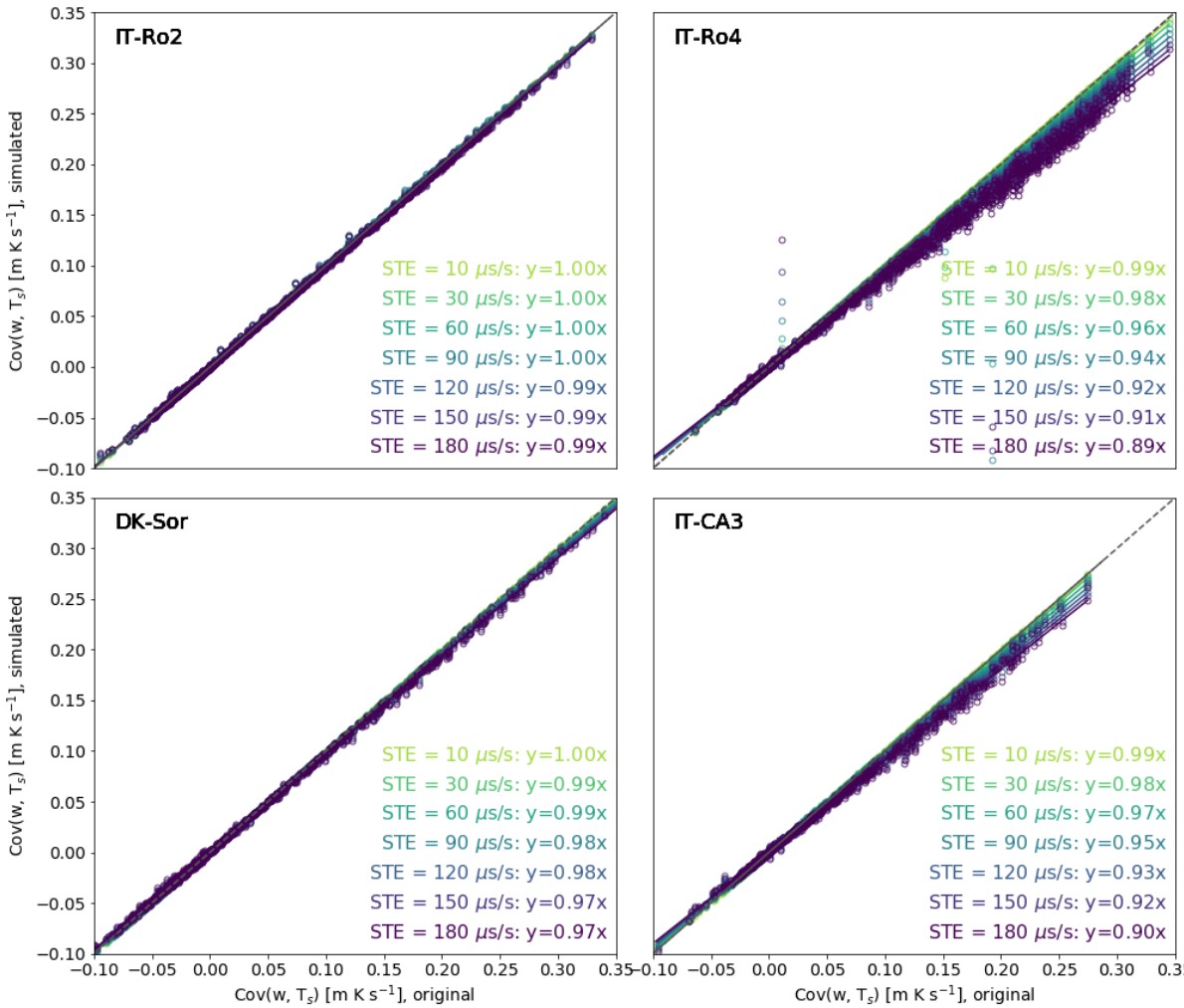

**Figure 5: Simulation of STE. Covariances *w-T_s* obtained with a set of simulated systematic errors of different amplitudes (y-axes) are compared to the covariances *w-T_s* computed with the original time series (x-axes), for the four sites. All regressions had offset equal to zero and r² > 0.98.**

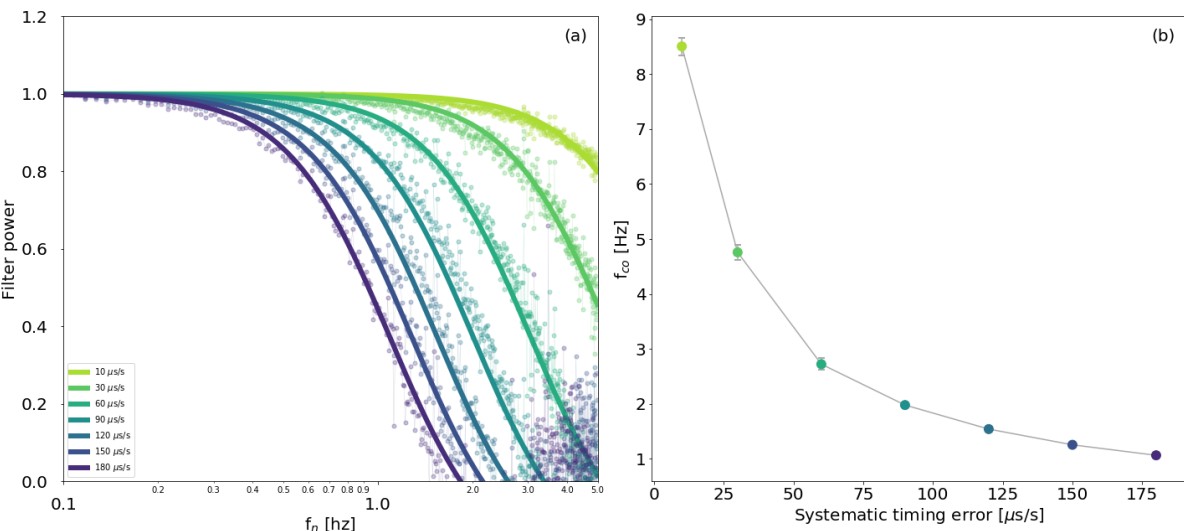

5  **Figure 6: Transfer function for the STE at different error amounts, as derived using Eqs. 2 and 3, using data from site IT-CA3 (a). Mean values and 3.5σ ranges of the transfer function cutoff frequencies across the fours sites, as a function of the error amount (b).**

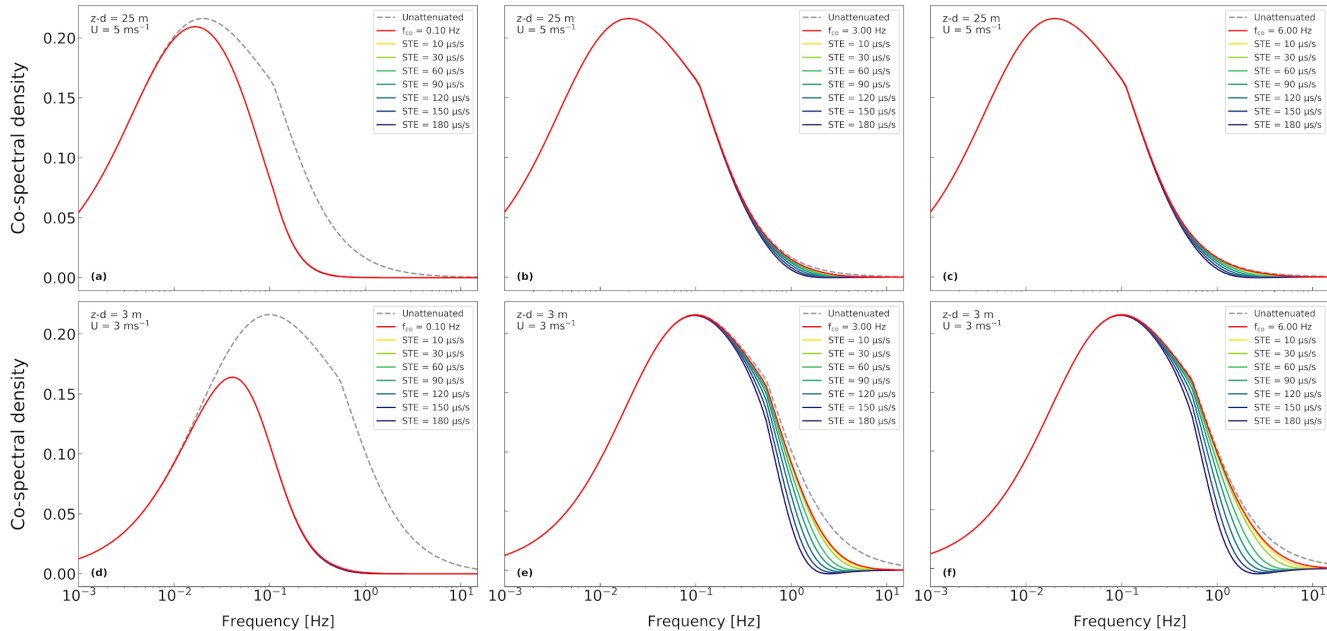

**Figure 7: Effect of adding artificial STEs to three EC systems characterized by different cutoff frequencies (0.1, 3.0 and 6.0 Hz, from left to right) and by different measurement height and mean wind speed (top to bottom).**

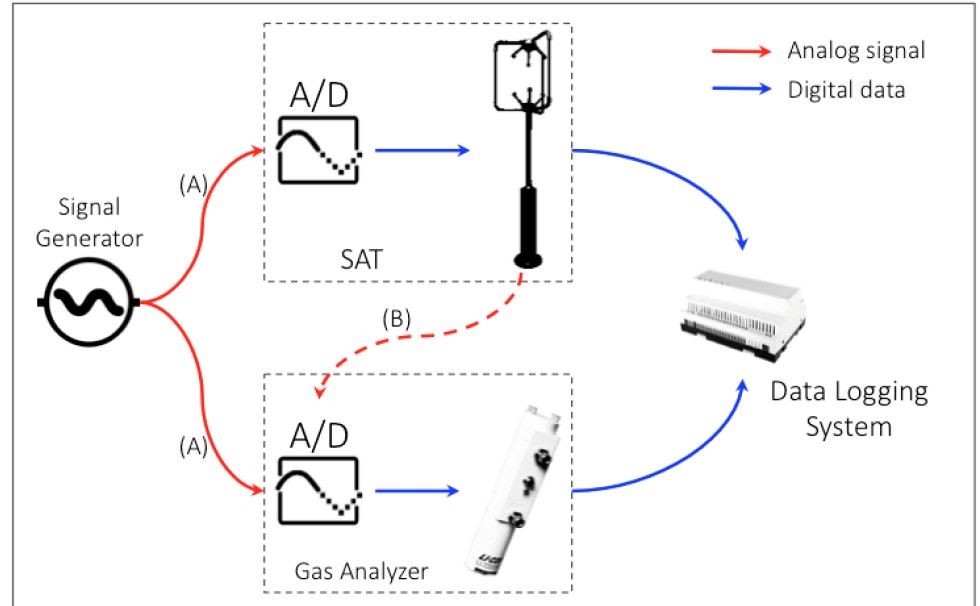

**Figure 8: Schematic of possible setups to evaluate the ability of a data logging system to synchronise EC data. In setup A the same (known) analog signal (solid red lines) is sent to the analog inputs of the EC instruments where it is digitized. In setup B analog wind data (dashed red line) is sent to the gas analyser, where it is digitized. In both setups, the 2 digital data streams are then collected by the data logging system. The clocks involved and how timestamps are attached to data depend on the specifics of the system under consideration.**

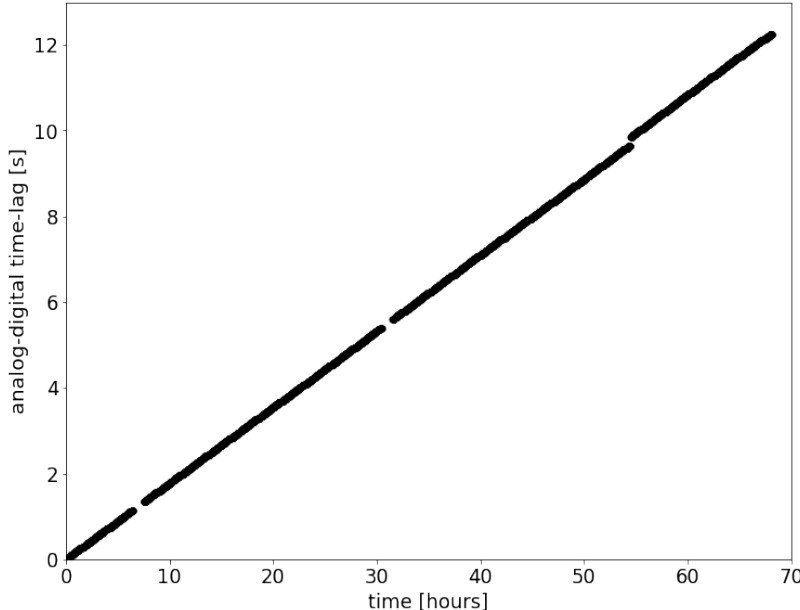

**Figure 9: Evolution of time-lags between two replicates of the same variable (*u* wind component, but identical results were obtained with *v* and *w*), one collected with the SAT native digital format and one collected via analog outputs from the SAT. Data were collected in two files roughly 70 hours long and then split into 30-min chunks for the computation of time-lags.**

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
