# Peer review of "Eddy Covariance flux errors due to random and systematic timing"

_Biogeosciences, 2018_

## Referee Comment (RC1) · Anonymous Referee #1 · 10 May 2018

This manuscript explores the effects of data stream synchronization on fluxes derived from all-digital data acquisition systems. While the methods and conclusions seem sound, there is little practical information about how this work applies to real-world systems such as TK3, EasyFlux, EdiSol, HuskerFlux, or SmartFlux. This work would be of greater value if the authors could review some of these systems and comment on whether or not the issues they explore are present or absent in any of these packages.

Never the less, I believe that this work does have value to the eddy covariance community and should be reported with several modifications.

1.) throughout the manuscript, "prospect" is used when "prospective" is appropriate 2.) on pg. 1, line 27, please define the term "zero-hold" 3.) on pg. 5, line 11, change "AT clocks" to "AT cut crystals".... also throughout the manuscript, please don't confuse

the term "clocks" (a system) with "crystals" (a component of a system). 4.) on pg. 6, line10, what do the authors mean by the term "vector"? A vector is a quantity that has magnitude and direction. How does this apply to time? 5.) on pg. 8, line 16, do the authors mean to use the term "filter" in this context? Does this imply that a mathematical operation was applied to the data in figure 5? 6.) on pg. 9, line 5, see #3 above

There may be other instances of undefined or confusing terms that I've missed. I'd encourage the authors to carefully review the manuscript for this.

Other more general comments follow:

The authors imply that all-digital data acquisition is a very recent development. This is not true. I've been aware of all-digital solutions for at least 15 years. One in particular (HuskerFlux from U. Nebraska or maybe Lawrence Berkeley Lab, I can't quite remember now) seems to have addressed a number of the issues identified here such as re-synchronization of data streams. The authors also imply that many of the synchronization problems outlined in the manuscript are absent from analog data acquisition systems, but this is not exactly true. Because of the "sample and hold" nature of A/D systems, many of these issues while present are masked. The authors also suggest that Ethernet connectivity is also relatively new, but again, this has been available for a long time, especially in Campbell Scientific data loggers (via the NL-100 module). One issue that the authors identify in serial data communications are the FIFO buffers used by many operating systems to ingest RS232 data. While these do exist and would create problems, well designed programs often get around this by lowering the size of these buffers and/or running independent program "threads" that handle individual character-received interrupts to pass the data along in near-real time. The authors also state that STE timing issues are not detectable, but I must disagree. When testing several data acquisition packages, I found that the HuskerFlux package recorded the individual buffer size differences after a user chosen interval. This difference can be used to calculate the magnitude of the STE over that interval. This should be relatively

easy for any new software to do. Finally, in developing a method to check any particular system for timing errors, the authors suggest using a signal generator to inject a single waveform into the A/D input of both instruments while having one instrument also send the same signal from it's D/A outputs to the second instrument. While this will work in principle, it must be cautioned that this is only strictly true if the D/A task and the A/D task are both synchronized with the measurement task and the serial output task in both instrument firmwares. Will this always be the case, or is this only true in some instruments such as the LiCor and Gill units tested by the authors? If these tasks are only loosely synched or are running asynchronously, then some issues could be masked by the internal asynchronicity.

I would encourage the authors to consider these issues and perhaps explore some of the available all-digital solutions to perhaps assist users in choosing a system for their purposes.

––––––––––––––––––––––––––

---

## Referee Comment (RC2) · Anonymous Referee #2 · 19 May 2018

General comments:

The manuscript titled, "Eddy Covariance flux errors due to random and systematic timing errors during data acquisition", by Fratini and co-authors describes an important, typically overlooked, source of error in eddy covariance resulting from timing issues and synchronization of EC instruments. The authors present a very clear and well-reasoned approach to quantify the potential magnitude of flux errors resulting from both random and systematic timing issues. Overall, the submission is sound and advances the state of EC methodology. However, a few suggestions to improve the manuscript are described below.

Specific comments:

[Figure]

1. The opening of introductory section was too abbreviated. Recommend additional description/background of the EC method including additional references. The definition of synchronicity could be improved. For example, an EC system with a fixed, known lag would be considered asynchronous by this definition but not one that leads to flux error in the context of this manuscript.

2. The manuscript has no explicit discussion of data triggering for digital data acquisition. This reviewer interpreted that all descriptions of digital data communications referred to streaming data. A brief discussion of triggering for data acquisition should be included in the Introduction, particularly as it relates to synchronizing data streams and timing errors.

3. The manuscript (section 1.2) describes open digital communication protocols including serial and Ethernet (packet-based) but do not address SDM (Synchronous Device for Measurements) communications. SDM is a very commonly used data communication protocol for collecting EC measurements and eliminates many of the timing errors described in the manuscript through clock synchronization. The authors should include a discussion of this protocol and which timing errors are applicable.

4. Considerable differences in flux errors (1 vs. 11%) were found between two sites given the same STE (180 $\mu$s/s). The explanation given was differences in the flux contribution in the frequency domain (cospectrum, see Figure 7) which is reasonable given the differences in observation height. In Fig 7, the cutoff frequency (transfer function) appears to differ between these two sites. However, in the text and as shown in Figure 6, the authors state that the transfer functions across sites were similar. Could this discrepancy be clarified?

5. One the main points made in the manuscript is that timing errors cannot be corrected or detected a posteriori. Given that the authors frame timing errors in the context of a low pass filter, it seems reasonable (assuming of spectral similarity between w'T' and w'c') that timing errors would be accounted for and corrected by spectral correction

methods that consider cospectra shape. Of course, such an approach could not differentiate between the source of signal loss (timing error, inlet tube attenuation, sensor separation, etc). The proposed approach assumes no timing error in the w'T' which is reasonable if calculated from a single SAT.

6. The manuscript would be strengthened if the findings were placed in the context of other sources of EC errors and uncertainties, particularly for fluxes of gas species. For example, one could apply the timing-error transfer function to the gas cospectra in concert with transfer functions of other spectral loses to illustrate relative contributions.

Technical corrections:

Page 1, Line 30: Consider using specifications in place of specs

Page 2, Line 17: Use consistent spelling of analyzer throughout manuscript

Page 2, Line 27: Replace others with other

Page 4, Line 39: Missing reference to (Hewlett Packard 1997)

Page 5, Line 21: Acronyms (TCP) should be spelled out prior to use.

Page 7, Line 2: Incorrect symbol to denote range

Page 8, Line 9: 30 minute (not minutes) file

Page 8, Line 10: Incorrect symbol to denote range

Page 8, Line 24: Aand should be and.

Page 9, Lines 19-20: use consistent spelling for serial protocol used; previously authors used RS-232.

Page 9, Line 23: Acronyms (OS) should be spelled out prior to use.

Page 10, Line 4: Native or naive?

Page 10, Line 12: Change to "thank".

Page 16: Clarify the figure and/or caption to denote that panel (a) illustrates transfer functions from a single site (IT-CA3).

Page 20, Line 32: Authors cite the discussion version of Langford et al., 2015. Consider the final version (doi:10.5194/amt-8-4197-2015)?

Page 21, Line 15: This reference (Smith 2002) was not cited in manuscript

---

## Short Comment (SC1) · 29 May 2018

The authors correctly refer to Eugster and Plüss (2010) where we argued that old-style traditional eddy covariance data acquisition systems used a combination of analog and digital data transmission. What the authors did not correctly reference is the fact that the Eugster and Plüss (2010) paper actually presents a high-quality fully digital data acquisition system and thus I am in disagreement with the authors in virtually all aspects of their manuscript. Fig. 1 below is a modified version of Fig. 1 in Eugster and Plüss (2010) showing the ideal fully digital acquisition system as Fig. 1c – this is what Eugster and Plüss (2010) recommend. The authors of this manuscript however try to put forward a downgrading of data quality that corresponds with Fig. 1b below. Although it is the authors freedom to have an opinion on this, their problems with their

sites cannot be generalised to other sites and namely their examples are not at all convincing me that downgrading a data acquisition which is using digital instruments to analog data transfer is yielding better flux results than keeping the digital data in digital format with as little loss as possible as suggested by Eugster and Plüss (2010).

On page 2, line 9 the authors of this manuscript write "Analog data output allows the data to easily cross clock domains. The clock that is used to sample the original signal does not need to be synchronized to the clock that samples the analog output. This makes it very convenient to merge data from systems with unsynchronized clocks." What they completely ignore is the fact that digital-to-analog conversion normally involves some stabilising R-C electronics (antialiasing) filter that dampens the original signal, and at the same time an analog-to-digital conversion is also equipped with a low-pass filter to avoid aliasing effects. Contrastingly, a fully digital data acquisition system of the kind proposed and used by Eugster and Plüss (2010) benefits from the same aspect claimed to be a quality of analog data acquisition: that only one clock is used for producing high-quality datasets. The concept used by Eugster and Plüss (2010) is to use the most reliable clock – that of the data acquisition computer that can be drift corrected with standard methods (in case Linux is used as an operating system, this can be achieved e.g. via the `/etc/adjtime` settings) or using some reliable time protocol services.

It is incorrect to say that "moving between clock domains is trivially simple in an analog system, it is much more challenging with digital data" (page 2, lines 31–32); this argumentation is simply ignoring that the authors use a one-clock system to collect data in the same way as Eugster and Plüss (2010) do it with a digital system, whereas analog input is treated clockless. It however reflects the authors personal skills to actually do this. For me it is as trivial to do digital synchronisation as these authors do analog synchronisation, but this has nothing to do with science but with personal experience; I have done it since 20 years, and hence I have full understanding that others find it less trivial.

On scientific grounds its however as trivial: with both analog and digital synchronisation you always want to merge the most recent measurement of one instrument with the most recent measurement of the other one. That's also the concept presented by Eugster and Plüss (2010): whenever a data record of the ultrasonic anemometer is received the most recent measurement of a gas analyser or fog droplet spectrometer available in the data queue is merged with the sonic anemometer data. But this is done in a fully digital mode (Fig. 1c), which means: the resulting dataset has the best possible date and time information of the Linux data acquisition system, uses the regular spacing of the reliable sonic anemometer (which has numbered records and thus any loss of data would be detected in a digital datastream), and at the same time each sonic record has the most recent information received from any additional analyser sending data in digital mode. The authors claim they have a better system but ignore that the only reason why an analog signal is present at any time at their analog input is because of some electronic (R-C) buffer that applies some degree of smoothing to that signal (which makes an analog signal valid over a longer timespan than a digital signal). With fully digital data acquisition of the type proposed by Eugster and Plüss (2010) (Fig. 1c) some rules have to be followed to avoid gaps in the data. Following the concept of analog data we suggest to simply repeat the previous record in case that no new one arrived from a gas analyser whenever the next (numbered) sonic record is received. Here, I would agree that improvements are possible since instruments have appeared on the market that use package-based non-realtime digital data transfer (TCP instead of UDP protocols, for example). The effect is, that if the data from the attached analyser are apparently arriving at a lower data rate than the nominal (and stable) data rate of the sonic anemometer, and if only the most recent arriving data record is retained and merged with sonic data. The potential effects of such problems was presented and discussed in quite some detail by Eugster and Plüss (2010). As a short summary: the effect on fluxes is small and well below empirical uncertainty of eddy covariance flux measurements (typically $\pm 10$–20%), but there is some damping introduced that needs to be corrected for. This is however also the case for a digital–analog–digital system

(Fig. 1b) due to the lowpass antialiasing filters used in the signal conversion in both directions (as depicted with "noise" in Fig. 1), and modern eddy covariance data processing software is capable of correcting such high-frequency damping losses. There is thus no scientific reason to believe that converting a digital signal to an analog one and then back again (Fig. 1b) will achieve better results than simply using the digital signal which can be processed without losses.

It is of course not impossible to generate a mismatch in timing that can lead to an underestimate in fluxes of up to 10% (page 2, line 14), but the authors forgot to mention that this is part of the high-frequency damping (see e.g. Eugster and Senn (1995)) that is corrected for in state-of-the-art systems. In Eugster and Plüss (2010) we have presented results how a mismatch of sampling frequencies of the sonic anemometer (running at nominally 20 Hz) and another analyser delivering data at 1/2 (10 Hz), 1/5 (4 Hz), or 1/15 (1.3 Hz) affect variances and fluxes. It is very clear that the better the data acquisition the lower this high-frequency loss correction, but that aspect is completely unrelated to the question whether analog or digital data acquisition is chosen, as long as we can agree that it is better to use one single clock (we use that of the Linux computer corrected for long-term drift) in combination with the very stable, continuous and lossfree data collection from the sonic anemometer (record numbers make sure this is lossless).

In their Fig. 9 the authors show that their clock has a drift of 12 seconds over 70 hours. This means, that their clock has a drift of 0.004762% – a ridiculously low value compared to the uncertainties of the single measurements performed by any sonic anemometer or gas analyser. In practice this means that if their drift means that the clock is too fast, then a system collecting at nominally 20.0 Hz is collecting at 20.00095 Hz, or if it is a slow clock this is 19.99905 Hz. Now, if we translate this effect to the accuracy of the flux, we add a scaling factor $\alpha$ to the nominal frequency $f$ expected

from an eddy flux data acquisition system. The reference flux with $\alpha = 1.0$ is

$$\overline{w'c'}_{ref} = \int\limits_{0}^{\infty} Co_{w,c}(f)df \ . \tag{1}$$

If frequency $f$ is slightly off by a factor $\alpha \approx 1.0$, then

$$\overline{w'c'}_{\alpha} = \int\limits_{0}^{\infty} Co_{w,c}(\alpha f)d(\alpha f) \tag{2}$$

$$= \alpha \int\limits_{0}^{\infty} Co_{w,c}(\alpha f)df \tag{3}$$

$$= \beta \, \overline{w'c'}_{ref} \ . \tag{4}$$

This means that the flux is enlarged by a factor $\beta \ll \alpha$. In Fig. 2 I simulated this effect using the parameterisation for normalised cospectra under neutral and instable conditions (Eq. 26 in Eugster and Senn (1995)). Thus, the flux – if only the drift of the main clock is of relevance – has to be multiplied with a factor $\beta$ that is much smaller than the drift. As a reading example: if $\alpha$ is unrealistically large with a value of 1.05 (i.e., 5% drift! See blue dashed lines in Fig. 2) then $\beta$ is on the order of 1.00015. In other words: even if the clock drifts by 5% then the flux will only be off by 0.015%, everything else being held constant. The reason is of course obvious: a 30-minute period over which we average remains a 30-minute period, even after correction for drift.

Thus drift is not the issue in this case. If there is jittering of the incoming data, then this will lead to a damping. The effect of damping was well addressed by Horst (1997) and before that by Eugster and Senn (1995), thus corrections for this effect exist, parametrisations exist, and these are actually applied in modern eddy covariance software.

The example shown by the authors in Figure 7 first indicate that the IT-Ro2 site has serious issues already in the original data. A correctly calculated cospectrum only shows such a change from the inertial subrange towards a white noise slope at the highest frequencies if there are serious issues with the sensors and/or data acquisition that lead to correlated noise in $w$ and $c$. If noise in $w$ and $c$ are uncorrelated as in a high quality data acquisition system (e.g. as presented by Eugster and Plüss (2010)), then such artefacts do not exist.

To illustrate this I downloaded some raw data collected yesterday (2018-05-28) by a high-quality ICOS Level 1 Candidate site (CH-DAV) that uses a data acquisition system of the type suggested by Eugster and Plüss (2010) (Fig. 1c). I used two hours of data from 13:30 to 15:30 CET and produced cospectra of the sensible heat flux (Fig. 3a) and the $CO_2$ flux (Fig. 3b). Neither of these shows any signs of white noise in the high frequencies. I chose a log-linear display since this is the only depiction that gives an optic representation of artefacts that is proportional to the area below the cospectrum. Thus, visually the integral under the bold curve is 1.0 (normalised cospectra), and any fraction of area at one frequency is the same size at another frequency. The blue dashed lines overlain over the cospectra are idealised undamped cospectra (see Eugster and Senn (1995) for equations and more details).

My experience as a reviewer is that often "cospectra" of the kind shown for IT-Ro2 in Fig. 7 are simply due to erroneous calculations of the cospectrum. I cannot double-check this hypothesis with the IT-Ro4 data shown at right in Fig. 7 since the authors hide the relevant part of the data at high frequencies in both panels, a bad practice irrespective of disagreements between the reader and the authors. Moreover, cospectra have both positive and negative cospectral densities (see my Fig. 3). If authors only show positive values it is unclear on wheather (a) they use the wrong calculation method, or (b) they hide negative values, or (c) they took the absolute values of the cospectral density. The method we used in Eugster and Plüss (2010) follows a concept that I learned from Ivan Mammarella, where different symbols are used for positive

and negative cospectral densities, and the absolute value is depicted. With such a display the scientific correctness is still fullfilled, whereas this manuscript suffers severely in all aspects.

Why did the authors not submit this to Atmospheric Measurement Techniques where editors are listed that have a much deeper understanding of such technical topics? Overall, I do not believe that this is sound science, in my view it is a huge step backwards, ignoring existing best available knowledge to a frightening degree, and hence I would fully support the Editor in his decision to reject this manuscript without suggestion to submit it elsewhere.

**References**

Eugster, W. and P. Plüss (2010) A fault-tolerant eddy covariance system for measuring $CH_4$ fluxes. *Agric. Forest Meteorol.* **150**, 841–851.

Eugster, W. and W. Senn (1995) A cospectral correction model for measurement of turbulent $NO_2$ flux. *Boundary-Layer Meteorol.* **74** (4), 321–340.

Horst, T. W. (1997) A Simple Formula For Attenuation of Eddy Fluxes Measured With First-Order-Response Scalar Sensors. *Boundary-Layer Meteorol.* **82**, 219–233.

[Figure]

*Signal smoothing and lightning protection filter

**Fig. 1.** Three variants of data acquisition systems. Modified after Eugster and Plüss (2010).

[Figure]

[Figure]

**Fig. 2.** The effect of a frequency that is off the nominal frequency by a factor $\alpha$ on the eddy flux.

[Figure]

**Fig. 3.** Example of two cospectra yesterday (2018-05-28) at CH-DAV, an ICOS Level I Candidate Site.

---

## Short Comment (SC2) · 1 Jun 2018

The paper draws the attention to an important topic in EC measurements sometimes inadvertently neglected by practitioners.

General comments:

The scope of the paper is too narrow and addresses only a certain class of eddy co-variance (EC) systems with instruments providing data to the data logging system via Ethernet or serial communication protocols and does not mention an important digital communication protocol, called Synchronous Devices for Measurement (SDM) that has been in use for more than 18 years. This protocol was specifically developed to meet the stringent requirements for time synchronization between the EC sensors. Two

sensor manufacturers, Licor Inc. and Campbell Scientific, Inc. collaborated and implemented this protocol in some of their instruments. The SDM protocol allows for external instrument triggering at precise moment in time, so multiple sensors can be measured synchronously. Some sonic anemometers, like the CSAT3 (Campbell Scientific, Inc.) can accept a trigger and provide almost instantaneous measurement when prompted by the datalogger. This approach does not require the complexity of high level of clock synchronization between individual devices.

The authors should include more details about the fundamental principle of operation of the devices used in EC systems and more specifically the widely used NDIR analyzers and sonic anemometers. Both of these sensors require some finite amount of time to make and process each measurement. They cannot provide a continuous analog signal, but rather discrete measurements. Consequently, the analog voltage outputs provided through the digital to analog converters (DAC) are discrete in time and magnitude.

A unique feature, specific only to the gas analyzer, is the use of a rotating optical filter wheel to multiplex the desired infrared bands used in the gas concentration measurements. The gas analyzer can produce a single measurement of $CO_2$ and $H_2O$ per each rotation of the filter wheel. This makes the frequency of the concentration measurements dependent on the rotational speed of the chopper wheel. If a precise timing is required the speed of the filter wheel need to be controlled precisely. Also, because of the dependency between the measurements and rotation of the wheel, the gas analyzer cannot be triggered to provide $CO_2$ and $H_2O$ readings at a precise moment in time. So, the only option to provide a reading at a given moment in time is to spin the filter wheel fast (like 150 rotations per second for the Li-7500) and report the measurement made immediately on the next rotation of the filter wheel. With this approach the gas readings could be at best synchronized to 1/150 second (6.7 ms), which is proven acceptable for most EC applications.

Similar approach should be implemented with sonic anemometers that do not have

a trigger mode and the ability to provide a measurement upon a request from the datalogging system.

A new generation of EC systems has been available for the last several years. These EC systems include a gas analyzer and a sonic anemometer, as co-located or as stand-alone devices, that share a common set of electronics so that wind, temperature and gas concentration measurements can be made simultaneously. This approach does not require precise clock synchronization and still provides sub millisecond timing between individual measurements.

The study examines only the errors in the co-variance of vertical wind and sonic temperature. The errors in the scalar fluxes could be strongly modulated by the density effects (WPL) associated with temperature and humidity. The scope of the paper can be extended to characterize the effect of timing errors on $H_2O$ and $CO_2$ fluxes which are of most interest in energy and carbon balance studies. The ability of the IRGASON to provide synchronous temperature, wind, $H_2O$ and $CO_2$ readings makes it a suitable instrument to study the reduction of co-variance not only on vertical wind and temperature, but on the other scalars. The implications of underestimated sensible heat flux due to systematic timing errors and its effects on the WPL terms and ultimately on the $CO_2$ flux could be addressed.

Additional information about the data sets used in the study should be included, like sample rate, sensor path length and anti-aliasing filter bandwidth.

The validation of the simulation design is not convincing, because the experimental conditions for the 100 Hz sonic data are unknown. (Co)spectral plots should be shown to verify the spectral content of the validation signals. It would have been more appropriate to use 100 Hz data from one of the EC sites.

Specific comments:

Page 1, Line 10: Synchronizing the clocks of the individual devices is only important

when the devices don't have a triggered mode.

Page 1, Lines 25-26: In the IRGASON the wind and the concentration data are made by the same instrument.

Page 1, Line 27 The separation between the sensors needs to be minimized. The requirement of the EC method is that the physical quantities are measured at the same point in space, so that co-variances are preserved

Page 2, Lines 9-10: Analog data output from a discrete sensor like a NDIR gas analyzer or a sonic anemometer can't cross clock domains. It can only be achieved, to some degree, if the measurement rate of the device is sufficiently high (above 100 Hz) and the digitization steps of analog signal becomes small.

Page 2, Line 21: raw measurements are not analog, but discrete, as was explained above.

Page 2, Line 25: Fully digital systems are not new. Sensors, like the CSAT3 and the Li7500 with SDM synchronous digital outputs, have been available for more than 18 years.

Page 4, Lines 8-11: This sentence is confusing and misleading. Synchronized wind and gas data is a requirement for the eddy covariance method. It can be achieved by co-locating the gas analyzer and the sonic anemometer, as in the IRGASON instrument. The spatial co-location eliminates any wind dependent time lags between the vertical wind and the sclalar of interest. The need for spectral corrections as proposed by Moore (1986), Horst (1997) and Horst and Lenschow (2009) is also eliminated. The spatial separation between sensors is a tradeoff between the spectral attenuation and the effects of flow distortion errors caused by the ultrasonic transducers, the supporting structure of the anemometer and the gas analyzers. Smaller diameter analyzers with horizontally symmetrical structure, as the EC150 and the IRGASON, are aerodynamic and can be positioned close to the sonic anemometer without causing significant

attenuation of vertical wind fluctuations, Fig. 4C Horst et al. (2016).

Page 4, Line 16: Reference for the Hydra-IV, CEH should be provided. I am not aware of any studies reporting flow distortion issues of the Hydra-IV, CEH. If such references exist, they should be provided or the Hydra-IV sensor should be removed from the list.

Page 6, Lines 17-18: Under certain conditions, flux biases could be drastically different from biases in covariances since the density effects (WPL) could strongly modulate the final flux calculations.

Page 7, Lines 27-28: Details about the conditions of the experiment should be provided. Spectra and co-spectra with 50Hz Nyquist frequency should be shown.

Page 10, Lines 7-10: SDM protocol should be mentioned as proven technological solution that guarantees sufficient synchronicity between sensors. There is a conflict between promoting collaboration between manufacturers and protecting technical ideas with patents. The authors should not recommend that the scientific community collaborate with manufacturers of EC equipment unless they are willing to share the ideas that they patented with United States Patent 9,759,703 B2. This patent claims protection of an IEEE adopted standard, Precision Time Protocol, for synchronization between eddy-flux instrumentation.

Page 17, Figure 7: The authors should explain the cluster of points at the high end of the frequencies for the time series with the 180 microseconds STE. It would be more appropriate to include Ogives plots to show if the two signals become out of phase at some frequencies causing negative values for the covariance.

REFERENCES:

Moore, C.J., 1986. Frequency response corrections for eddy correlation systems. Boundary-Layer Meteorology, 37: 17-35

Horst, T. W.: A simple formula for attenuation of eddy fluxes measured with first-order-response scalar sensors, Bound.-Lay. Meteorol., 82, 219–233, 1997.

Horst, T. W. and Lenschow, D. H.: Attenuation of Scalar Fluxes Measured with Spatially-displaced Sensors, Bound.-Lay. Meteorol., 130, 275–300, 2009.

United States Patent 9,759,703 B2. Systems and Methods for Measuring Gas Flux

---

## Author Comment (AC1) · 20 Jun 2018

**Reply to Short Comment by W. Eugster**

W. Eugster (hereafter WEug) puts forward a number of criticisms, which can be summarized in the following 3 points:

1. WEug criticizes the fact that we supposedly suggest to use mixed analog-digital acquisition systems instead of fully digital systems. The first 4 pages of the comment elaborate on this topic using a data acquisition system that he proposed in the past as a contrasting example.
2. At pages C4-C5 WEug proposes a mathematical demonstration as well as a simulation and a practical example, supposedly showing that "drift [*which we identify as the major source of flux bias*] is not the issue".
3. At pages C5-C7 WEug criticizes our Fig. 7 because, in his opinion, it depicts (possibly wrongly calculated) co-spectra in the wrong manner.

In the last paragraph, WEug criticizes our choice of the Journal and proposes to reject the paper.

Following our replies to each point:

1. WEug misunderstood the message we want deliver with this paper. By no means do we suggest using mixed analog-digital acquisition systems. We merely start from the observation that in traditional analog-digital EC systems, data synchronization was not an issue the researcher needed to be too much concerned of, because the instruments' manufacturers essentially solved it and methods exist for correcting for the (potentially) involved spectral losses in post-processing. Conversely, when assembling a custom fully digital system – which is often necessary in applications, for which industrial-grade integrated systems do not yet exist – care must be taken to get the synchronization right. Therefore, the entirety of the paper is devoted to describing and demonstrating (via simulations starting from real data) potential flux biases resulting from poorly designed fully digital acquisition. However, several times we clearly state that fully digital systems are to be preferred. As an example, at Pag. 2, Lines 12-25 we highlight the limits of analog-digital systems and the following Lines 26-30 explain how fully digital acquisition overcomes those limits. Furthermore, the opening line of the Conclusions reads: "Undoubtedly, modern EC systems should log high-frequency data in a native digital format […]". It appears that Referee #1 and #2 - as well as Ivan Bogoev in his Short Comment - didn't misunderstand this crucial point, so we don't deem it necessary to modify the paper for this aspect, unless the Editor requires it.

2. WEug misunderstood the drift issue we describe and simulate. The entire paper discusses *relative* frequency drifts, i.e. the drift of the clock of the sonic relative to the clock of the gas analyzer (or vice-versa). He instead describes, simulates and exemplifies an *identical* absolute drift of the two clocks, which of course results in no *relative* drift and no significant errors. This is evident from both Eq. 2 of his Short Comment and from the discussion that follows it. Again, the Referees and the author of the second Short Comment appear to have understood what we have done and its rationale, so we don't deem it necessary to make any modifications, unless prompted by the Editor.

3. WEug misunderstood what Fig. 7 shows and therefore derived irrelevant conclusions. We do appreciate, though, that the description of the figure could be improved. For that figure, we used data from 2 sonic anemometers and computed co-spectra between vertical wind component ($w$) and sonic temperature ($T_s$), the grey circles in Fig. 7. We then modified $T_s$ to simulate a *relative* drift and computed co-spectra again, the purple circles. The aim is to show the low-pass nature of the filter that the relative drift implies and how the same filter leads to significantly different losses depending on the co-spectral shapes. We note that:

   - No data from a gas analyzer is involved, so considerations about correlated noise etc. (beginning of Pag. C6 of the Short Comment) don't apply.
   - In our opinion, a log-log plot is better suited to visualize spectral losses at high frequencies (which is the aim of Fig. 7) because it emphasizes the lower y-axis ranges, where attenuation occurs. However, due to the stark difference in attenuation, a semi-log plot works as well in this instance - see the following figure, where we plotted the same data on a linear y-axis:

[Figure]

   Here we also expanded the axis ranges to show the entirety of the data. In this version of the Figure one can see that at high frequency the transfer function is characterized by damped oscillations around zero (the same can be guessed by looking at Fig. 6a of the manuscript,

bottom-right corner). This is an aspect we did *not* want to emphasize in the paper, because we believe it has no practical implications (oscillations occur where signal is already minimal) and only distracts from the main message of the paper. This is also the reason why at Pag. 7, lines 1-2 we state that we used a transfer function model "that was found to reasonably approximate the data obtained for all drifts at all sites *in the most relevant frequency range*". The model captures the transfer function behavior well at frequencies where co-spectral content is significant and, in particular, where the cut-off frequency is found.

In any case, addressing a comment by Referee #2, we suggest to replace this Figure with the one proposed in the Reply to Referee #2, Comment 6.

In addition, WEug writes: "*My experience as a reviewer is that often "cospectra" of the kind shown for IT-Ro2 in Fig. 7 are simply due to erroneous calculations of the cospectrum. I cannot double check this hypothesis with the IT-Ro4 data shown at right in Fig. 7".* We need to point out that, upon his request, on April 17, 2018 we shared with WEug, by email, the data and the code used to generate Fig. 7 (with no response from him until the publication of his Short Comment), therefore he had with him everything needed to "*double check [his] hypothesis*".

Finally, we find the suggestion to submit to a different Journal somewhat unusual, considering that the Editor already evaluated the paper to be appropriate for publication in Biogeosciences Discussion and that WEug posted his Short Comment after the Referees have submitted their reviews, which show a full understanding of the discussed issues and of the content of our manuscript.

---

## Author Comment (AC2) · 20 Jun 2018

**Reply to Short Comment by Ivan Bogoev**

We thank Ivan Bogoev (IB hereafter) for his consideration of our manuscript, although we didn't particularly appreciate the repeated reference to instrumentations and protocols developed and commercialized by IB's company, in a context where no specific instrumentation is discussed and without strong relevance to the topics discussed in the manuscript.

Following are our replies, breaking down the text into individual comments.

*Comment 1:*
*The scope of the paper is too narrow and addresses only a certain class of eddy covariance (EC) systems with instruments providing data to the data logging system via Ethernet or serial communication protocols and does not mention an important digital communication protocol, called Synchronous Devices for Measurement (SDM) that has been in use for more than 18 years. This protocol was specifically developed to meet the stringent requirements for time synchronization between the EC sensors. Two sensor manufacturers, Licor Inc. and Campbell Scientific, Inc. collaborated and implemented this protocol in some of their instruments. The SDM protocol allows for external instrument triggering at precise moment in time, so multiple sensors can be measured synchronously. Some sonic anemometers, like the CSAT3 (Campbell Scientific, Inc.) can accept a trigger and provide almost instantaneous measurement when prompted by the datalogger. This approach does not require the complexity of high level of clock synchronization between individual devices.*

In our manuscript we intentionally avoided to mention and discuss existing commercial solutions (including the SmartFlux™ system released by LI-COR Biosciences, to which two of the Authors are affiliated). Rather, we focused our attention on the description and quantification of the timing-related errors potentially involved in fully-digital acquisition system, and on protocols and solutions that are scalable and likely to be applicable to a wide range of instrumentations and gas species. It is correct to say that a data collection strategy based on instrument triggering would avoid STEs (though not necessarily RTEs). However, implementing a triggering-based system is much more complicated - when even possible at all - for typical research groups that design and realize their own EC acquisition system. For example, to the best of our knowledge most sonic anemometers and gas analysers (also beyond $CO_2$ and $H_2O$) do not support a triggering signal. In particular, SDM is a Campbell Scientific Inc. proprietary protocol, implemented in just a handful of instruments (manufactured by only LI-COR Biosciences and Campbell Scientific) and very unlikely to ever be implemented in new instrumentation, other than Campbell's. For these reasons, we consider SDM a commercial solution of interest in specific cases, but of little relevance in general, for future EC systems, and for the scope of our manuscript.
Nonetheless, prompted also by Referee #2, in the revised manuscript we will add a note on SDM and discuss triggering-based systems (see also Replies to Referee #2).

*Comment 2:*
*The authors should include more details about the fundamental principle of operation of the devices used in EC systems and more specifically the widely used NDIR analyzers and sonic anemometers.*
*Both of these sensors require some finite amount of time to make and process each measurement. They cannot provide a continuous analog signal, but rather discrete measurements. Consequently, the analog voltage outputs provided through the digital to analog converters (DAC) are discrete in time and magnitude. A unique feature, specific only to the gas analyzer, is the use of a rotating optical filter wheel to multiplex the desired infrared bands used in the gas concentration measurements. The gas analyzer can produce a single measurement of CO2 and H2O per each rotation of the filter wheel. This makes the frequency of the concentration measurements dependent on the rotational speed of the chopper wheel. If a precise timing is required the speed of the filter wheel need to be controlled precisely. Also, because of the dependency between the measurements and rotation of the wheel, the gas analyzer cannot be triggered to provide CO2 and H2O readings at a precise moment in time. So, the only option to provide a reading at a given moment in time is to spin the filter wheel fast (like 150 rotations per second for the Li-7500) and report the measurement made immediately on the next rotation of the filter wheel. With this approach the gas readings could be at best synchronized to 1/150 second (6.7 ms), which is proven acceptable for most EC applications. Similar approach should be implemented with sonic anemometers that do not have a trigger mode and the ability to provide a measurement upon a request from the data logging system.*

We agree with this comment and thank IB for the suggestion. We will elaborate on this concept in the Introduction of the revised manuscript. We note, however, that the uncertainty deriving from the time-discrete nature of the measurement is random and does not accrue with time. Therefore, it can be assimilated to an RTE, whose effect on fluxes we verified to be negligible, as long as the precision of the measurement cycle timing is sufficiently smaller than the sampling frequency. This is certainly the case in NDIR-based analyzers.

*Comment 3:*
*A new generation of EC systems has been available for the last several years. These EC systems include a gas analyzer and a sonic anemometer, as co-located or as stand-alone devices, that share a common set of electronics so that wind, temperature and gas concentration measurements can be made simultaneously. This approach does not require precise clock synchronization and still provides sub millisecond timing between individual measurements.*

As already mentioned in the original manuscript, "*Commercial solutions implementing sound engineering practices do exist for long-established EC of $CO_2$ and $H_2O$, which guarantee data synchronicity within specifications that meet EC requirements.*" [Pag. 1, Lines 29-30]. The ones referred to by IB (implicitly here,

and explicitly several times later) are just a few among those. Our manuscript is concerned primarily with EC systems assembled by research groups or individuals, to address needs which are either not yet met by industrial partners, or are met with solutions not scalable to other existing EC instrumentation.

*Comment 4:*
*The study examines only the errors in the covariance of vertical wind and sonic temperature. The errors in the scalar fluxes could be strongly modulated by the density effects (WPL) associated with temperature and humidity.*

While we indeed used sonic temperature in our simulations, the intention was not that of estimating errors in "covariance of vertical wind and sonic temperature". A sonic anemometer always measures sonic temperature, too, and therefore there is no timing issue there. The simulation using modified sonic temperature is intended to describe what happens to a scalar measured by a *separate* instrument under various scenarios of RTEs and STEs.

Of course, WPL effects modulate fluxes. If timing issues affect covariances between $w$ and $c$, everything downstream is affected, including the $H_2O$-related WPL terms (but not the $T_s$-related terms). Resulting errors could be somewhat smaller or larger, depending on factors such as the relative error in $w$-$c$ covariance and the magnitude of the dilution-related WPL terms relative to the uncorrected flux.

*Comment 5:*
*The scope of the paper can be extended to characterize the effect of timing errors on H2O and CO2 fluxes which are of most interest in energy and carbon balance studies. The ability of the IRGASON to provide synchronous temperature, wind, H2O and CO2 readings makes it a suitable instrument to study the reduction of co-variance not only on vertical wind and temperature, but on the other scalars. The implications of underestimated sensible heat flux due to systematic timing errors and its effects on the WPL terms and ultimately on the CO2 flux could be addressed.*

As explained in the reply to the previous comment, our manuscript is *already* concerned with gas fluxes. Considering all other effects playing a role in the flux computation (flow distortion, for an example) is outside the scope and in the specific case would actually *reduce* the scope by concentrating on the specifics of $CO_2/H_2O$, while the study is agnostic to the gas species under observation and only relies on the spectral similarity assumption.

This is important, in that we expect synchronization issues to be a significant concern in, for example, $CH_4$ and $N_2O$ flux measurements.

*Comment 6*
*Additional information about the data sets used in the study should be included, like sample rate, sensor path length and anti-aliasing filter bandwidth.*

We will consider adding this information in the revision of the manuscript, if proved to provide any additional insights.

*Comment 7*
*The validation of the simulation design is not convincing, because the experimental conditions for the 100 Hz sonic data are unknown. (Co)spectral plots should be shown to verify the spectral content of the validation signals. It would have been more appropriate to use 100 Hz data from one of the EC sites.*

Given the virtually identical results obtained using 10 or 100 Hz data, we think that elaborating on experimental conditions and adding plots and details on the validation procedure detracts from the clarity and straightforwardness of the message, without adding anything useful. Though not stated explicitly (and we can fix that in the revision), the 100 Hz data was collected under conditions representative of typical EC measurements. As IB is probably aware of, 100 Hz turbulence data are not straightforwardly available, especially to prospective readers of our manuscript (EC practitioners). We wanted to setup a simulation that could easily be replicated by others, using also the source code that we plan to make available. For this reason we used 10/20 Hz data, and only used 100 Hz data to validate the procedure.

**Other comments**

IB added a number of more specific comments referring to specific pages and lines. Most of them are related to the points already discussed above, so we consider them answered and clarified.
However there is one comment that we would like to address. IB wrote: *"There is a conflict between promoting collaboration between manufacturers and protecting technical ideas with patents. The authors should not recommend that the scientific community collaborate with manufacturers of EC equipment unless they are willing to share the ideas that they patented with United States Patent 9,759,703 B2. This patent claims protection of an IEEE adopted standard, Precision Time Protocol, for synchronization between eddy-flux instrumentation."*

Firstly, we need to point out that the Authors of the manuscript are from different Institutions - not only from LI-COR Biosciences - and therefore they collectively don't own any patent. We urge more rigour and respect for the role and work of each individual.
The statement *"There is a conflict between promoting collaboration between manufacturers and protecting technical ideas with patents"* is factually incorrect. The patent system is designed to protect the inventor from infringement, and it does not impede collaboration (please refer to e.g. GitHub or any public software repository for an example). In fact, we argue that this manuscript and the patent referenced by IB *are exactly* a way of sharing ideas with the academic research community, while protecting intellectual property.

To the point of the manuscript under discussion, the Authors, and in particular G.F and K.E. of LI-COR Biosciences, think that using open and widely adopted protocols in conjunction with proprietary firmware to implement scalable and "agnostic" data acquisition systems strikes the perfect balance between: (1) allowing flexibility while guaranteeing high-quality measurements and (2) protecting commercial interest.

Finally, IB points out that:

*Page 17, Figure 7: The authors should explain the cluster of points at the high end of the frequencies for the time series with the 180 microseconds STE. It would be more appropriate to include Ogives plots to show if the two signals become out of phase at some frequencies causing negative values for the covariance.*

We addressed this point in the context of a reply to the Short Comment by W. Eugster (see Point 3, second bullet in that reply).

---

## Author Comment (AC3) · 20 Jun 2018

**Reply to Referee #1**

We would like to thank the Referee for a useful review and for the suggestions of further interesting work related to our manuscript. In the following, we break down the Referee's text and address each comment individually.

*Comment 1:*

... *[ ] there is little practical information about how this work applies to real-world systems such as TK3, EasyFlux, EdiSol, HuskerFlux, or SmartFlux. This work would be of greater value if the authors could review some of these systems and comment on whether or not the issues they explore are present or absent in any of these packages*

This is an important point that we also considered while preparing the manuscript. With this article, we wanted to present the problem of synchronization in fully-digital systems and characterize and quantify the corresponding errors. The aim was to reach out to developers of acquisition systems (industrial- or research-grade) as well as to EC practitioners in order to highlight the importance of this issue and its impact.

In addition, synchronization issues are relevant not only to $CO_2$ and $H_2O$ but also to other gases. In fact, it can be presumed that they are even more relevant for gas species that to date received less industrial investment in terms of system integration, with the result that usually the data acquisition system must be designed and assembled by the researcher, given what is made available by the instruments' manufacturers.

For this reason, instead of undertaking an analysis of the compliance for each acquisitions system today available or a quantification of its errors - which would have been extremely long, time consuming, incomplete and soon outdated - we focussed the manuscript on the description of the issue itself and propose a simple test for the evaluation of each given system. It can be noted, in fact, that even the acquisition system developed and commercialized by the company of two of the coauthors (SmartFlux™ by LI-COR Biosciences) is never cited in the paper.

In addition, since some of the solutions mentioned are commercial products, their evaluation could only be undertaken by involving all involved parties. This is something certainly outside the scope of our work, but our hope is that this article will now enable such an analysis, providing a reference framework.

*Comment 2:*

*One issue that the authors identify in serial data communications are the FIFO buffers used by many operating systems to ingest RS232 data. While these do exist and would create problems, well designed programs often get around this by lowering the size of these buffers and/or running independent program "threads" that handle individual character-received interrupts to pass the data along in near-real time*

We do agree with the Referee that "well designed systems" can avoid the problems we present. In the Introduction, we also point out that "commercial solution exist […]". In the revised manuscript, we will add and better clarify that well engineered solutions also exist and can be developed (by both commercial companies and non-commercial institutions), provided the appropriate engineering skills (electronics, computer science, digital signal processing, etc.) are available.

*Comment 3:*
*The authors also imply that many of the synchronization problems outlined in the manuscript are absent from analog data acquisition systems, but this is not exactly true. Because of the "sample and hold" nature of A/D systems, many of these issues while present are masked.*

The Referee is correct. The measurement principle of the instruments covered by this paper is generally discrete-time (i.e. there is a defined "measurement interval" in each instrument). There is, in fact, an unavoidable zero order hold in any analog output from such an instrument, which will be governed by the clock of the source instrument. When subsequently sampled by a "sample and hold" A/D system running on an independent clock, the sampling error that will occur will be of random nature (RTEs), and inversely proportional to the output rate of the analog signal (i.e. the higher the output rate, the smaller the resulting RTE). We will add this and correct our statement in the new version of the manuscript.

*Comment 4:*
*Finally, in developing a method to check any particular system for timing errors, the authors suggest using a signal generator to inject a single waveform into the A/D input of both instruments while having one instrument also send the same signal from it's D/A outputs to the second instrument. While this will work in principle, it must be cautioned that this is only strictly true if the D/A task and the A/D task are both synchronized with the measurement task and the serial output task in both instrument firmware. Will this always be the case, or is this only true in some instruments such as the LiCor and Gill units tested by the authors? If these tasks are only loosely synched or are running asynchronously, then some issues could be masked by the internal asynchronicity.*

It would seem that the Referee partially misunderstood the proposed test: "using the signal generator" and "using the analog signal of one instrument" are proposed as two *alternative* ways of reaching the same objective, they don't need to occur together. The important point in evaluating synchronization performance is to craft a means of producing a known signal, sampled independently by the two clocked systems. The resultant dataset, as captured by the proposed sampling system, can be directly evaluated, as the correlation should be perfect under these conditions. Nonetheless, the raised concern on the first solution in particular (using the signal generator) holds true and may indeed limit applicability of the test to other instrumentation and to teams inexpert in data acquisition systems design. In the revised manuscript we will add a cautionary note highlighting this point .

*Minor Comments:*

We will modify the manuscript to accommodate the Referee's comments and suggestions. In particular, we will de-emphasize the "novelty" of fully-digital data acquisition in EC, which are indeed not so "new" as we initially presented them, although the widespread use of digital acquisition in the EC community is relatively new.

---

## Author Comment (AC4) · 20 Jun 2018

**Reply to Referee #2**

Firstly, we would like to thank the Referee for a useful review that we believe can help us improve the manuscript significantly. Following are replies to the Referee's comments.

*Comment 1:*
*The opening of introductory section was too abbreviated. Recommend additional description/background of the EC method including additional references. The definition of synchronicity could be improved. For example, an EC system with a fixed, known lag would be considered asynchronous by this definition but not one that leads to flux error in the context of this manuscript*

We agree and are now extending the introduction with a richer description of the Eddy Covariance method, and reviewing the definition of synchronicity. We would like to note, however, that - for the sake of completeness - we did intend to include time lags among the misalignments, only to later ignore it because it is a *correctable* misalignment. For this reason, the revised definition of "synchronous" would still not include an EC system with a (fixed) time lag.

*Comment 2:*
*The manuscript has no explicit discussion of data triggering for digital data acquisition. This reviewer interpreted that all descriptions of digital data communications referred to streaming data. A brief discussion of triggering for data acquisition should be included in the Introduction, particularly as it relates to synchronizing data streams and timing errors.*

The Referee correctly interpreted that the paper is concerned with systems based on streamed data. As suggested, we added a brief discussion of systems based on data triggering and clarified that we concentrate on streaming-based systems. The reason for this choice is that implementing a triggering-based system is much more complicated to the non-specialist and it is very often just not allowed by the instruments. For example, to the best of our knowledge most sonic anemometers and gas analysers (also beyond $CO_2$ and $H_2O$) do not support a triggering signal, while all instruments we are aware of do provide data in streaming mode.

*Comment 3:*
*The manuscript (section 1.2) describes open digital communication protocols including serial and Ethernet (packet-based) but do not address SDM (Synchronous Device for Measurements) communications. SDM is a very commonly used data communication protocol for collecting EC measurements and eliminates many of the timing errors described in the manuscript through clock synchronization. The authors should include a discussion of this protocol and which timing errors are applicable*

SDM does not "synchronize clocks", but rather implements a triggering strategy to avoid STEs (while still being subject to potential RTEs). It is therefore an

instance of a system based on triggering signals, that will be discussed in the new version of the manuscript as described above. We will also briefly reference the SDM protocol, which, however, is a Campbell Scientific Inc. proprietary protocol, implemented in a small number of instruments (manufactured by only LI-COR Biosciences and Campbell Scientific) and very unlikely to ever be implemented in new instrumentation, other than Campbell's. We have designed our manuscript without explicit reference to specific commercial solutions (including SmartFlux™, released by LI-COR Biosciences) and we therefore intend to only add a note on SDM, without entering in any details.

*Comment 4:*
*Considerable differences in flux errors (1 vs. 11%) were found between two sites given the same STE (180 µs/s). The explanation given was differences in the flux contribution in the frequency domain (cospectrum, see Figure 7) which is reasonable given the differences in observation height. In Fig 7, the cutoff frequency (transfer function) appears to differ between these two sites. However, in the text and as shown in Figure 6, the authors state that the transfer functions across sites were similar. Could this discrepancy be clarified?*

While we agree with the Referee and have been puzzled as well by the visually perceived difference between the two transfer functions implied in Fig. 7, we came to believe that the plot correctly shows the effect of the *same* transfer function on two cospectra with very different shapes. In the reply to Comment 6 (see later) we propose a new Figure that makes use of model cospectra, to provide more details about how and when STEs generate significant errors in the form of spectral losses. In that Figure, the same visual effect of Fig. 7 can be seen: compare, for example, plots (c) and (f). We do suggest to replace current Figure 7 with the figure below, which allows more elaborate discussion.

*Comment 5:*
*One the main points made in the manuscript is that timing errors cannot be corrected or detected a posteriori. Given that the authors frame timing errors in the context of a low pass filter, it seems reasonable (assuming of spectral similarity between w'T' and w'c') that timing errors would be accounted for and corrected by spectral correction methods that consider cospectra shape. Of course, such an approach could not differentiate between the source of signal loss (timing error, inlet tube attenuation, sensor separation, etc). The proposed approach assumes no timing error in the w'T' which is reasonable if calculated from a single SAT*

We agree with the Referee that, in principle, an *in-situ* spectral correction method based on co-spectra would indeed correct timing errors as well. One caveat is that, in our opinion, spectral attenuations in the gas analyzer are better assessed using solely gas spectra rather than co-spectra (for the reasons put forward in [1]). It also appears to us that addressing the different sources of spectral losses individually allows for more control and fine-tuning of the correction.

We will modify the discussion to better clarify this point and suggest that, at least in principle, a spectral correction based on cospectra would indeed correct for STEs. That said, we would also stress that one should always strive to avoid eliminable sources of bias errors, and STEs/RTEs fall in this category.

*Comment 6:*
*The manuscript would be strengthened if the findings were placed in the context of other sources of EC errors and uncertainties, particularly for fluxes of gas species. For example, one could apply the timing-error transfer function to the gas cospectra in concert with transfer functions of other spectral loses to illustrate relative contributions*

In the original manuscript (Pag. 8, Lines 22-27) we tried to put STE-induced errors in perspective by comparing the corresponding cut-off frequencies with those found in literature for modern $CO_2/H_2O$ EC systems. In general, we think that discussion of spectral losses of any given EC system should be limited to transfer functions and cut-off frequencies, because these are properties of the system itself, regardless of its deployment. Actual flux errors and relative contribution of different sources of spectral losses, as well as the other sources of uncertainty and error, depend critically on measurement height, turbulence regime and site characteristics.

[Figure]

**Figure AC1. Effect of adding different STE to 2 EC systems characterized by different cut-off frequencies (left to right) and by different measurement height and mean wind speed (top to bottom). It is evidenced that: at high measurement heights effects are negligible irrespective of the "original" cut-off frequency of the system (a-c); at low measurement height, STEs significantly increase spectral losses if the system has a high "original" cut-off frequency (e-f). If the system as a poor spectral response to start with, STEs are irrelevant (d).**

Nonetheless, following the suggestion of the Referee, we prepared Figure AC1, which could be added to the discussion for a quantitative, hopefully more intuitive, understanding of the errors potentially caused by STEs. Further, we suggest that Figure AC1 could replace current Figure 7, as explained above (see reply to Comment 4).

The figure shows how different STEs would affect the overall transfer function of a given EC system affected by other sources of spectral losses.

*Technical Corrections:*

In the revised manuscript, we will accept and implement all minor editing suggestions of the Referee.

**References**

[1] Ibrom, A., Dellwik, E., Flyvbjerg, H., Jensen, N.O., Pilegaard, K., 2007. Strong low-pass filtering effects on water vapour flux measurements with closed-path eddy correlation systems. Agric. For. Meterol. 147, 140–156.

---

## Author Response (AR1)

**Detailed answers and modifications to manuscript**

Please find below the point-by-point answer to the Editor and Referees comments (answers in blue). Note that in this document we didn't report the concepts and discussions that we already put in the answers to the reviewers and short comments (already published in the Discussion) but rather list what we in fact did and why, for each of the points.

**Replies to the Associate Editor**

*A common theme across the reviews is that the manuscript is somewhat misleading in terms of the extent of the problem and overlooks the long-history of efforts to address it. The reader comes away with the impression that timing errors of the magnitude reported here could affect the majority of eddy-covariance records, and a few of the reviews had the impression that you were advocating analog-only solutions.*

In our understanding, only one reader (not a Referee) had the impression that we were advocating for analog-only or mixed analog-digital solutions. We think that we made it sufficiently clear that fully-digital systems are to be preferred, nevertheless we significantly modified the manuscript in an attempt to avoid any risks of misinterpretation in that sense.

*You should make very clear throughout the manuscript (especially abstract, intro, discussion and conclusions) that your work applies only to in-house developed EC systems. It would also be great if you could give a sense of what proportion of EC systems are in-house. I understand this is difficult, but it is important to give the reader a sense of whether the vast majority of systems are commercial, or in-house.*

We share the Editor's opinion, that our concerns apply mainly to "in-house" systems. However, we cannot exclude that timing errors do exist also in commercial solutions. In the revised manuscript, and notably in the largely rewritten Section 1.2, we elaborate on what an user could realistically expect from systems of different types.

We do not think we have enough information to speculate on the fraction of "in-house" fully-digital systems, because the relevant metadata are extremely scarce even for large networks such as AmeriFlux and EuroFlux, but we did add a paragraph on this in the conclusions.

*I understand you want to exclude commercial solutions, but I do not think that makes sense. You are highlighting a problem to which a solution has been developed, and it would be much more helpful to the community to direct readers to those solutions. You could however use the magnitude of the problem to call on commercial organizations to open source their solutions, or at the very least to highlight that the*

*degree of implementation across commercial platforms is not quantifiable (as you claim). This lack of an*
*acknowledgement of existing solutions came up a lot with the reviewers.*

We thoroughly revised the manuscript to acknowledge existing solutions, both commercial and from the community, mainly in the Introduction and in the Conclusions. At the same time we constrained the scope of our paper to mainly "in-house" systems operating in streaming mode. We do not think we have enough information and/or experience with many of the existing solutions to elaborate in details on their synchronization capabilities, so we limited our discussion to the general features of such solutions. This is also not the aim of our paper anyway.

*You mention that most solutions are not scalable to other species than CO2 and H2O. What about CH4,*
*for which Eugster and Plüss (2010) present a fault tolerant system? The measurement of other species*
*(e.g., H2, N20, O3, BVOCS), remains very rare across global networks, which brings us back to the*
*question of how widespread the issue of timing errors is.*

Many solutions - notably those based on triggering or polling systems - are indeed not scalable to instrumentation (either anemometers or gas analyzers) other than those for which they are designed, because they require two-way communication between the logging system and the instrument, that needs to be implemented in the instrument's firmware. Solutions relying on data transmission in streaming mode are more scalable, but the details matter critically, and it is therefore impossible for us to make an assessment of the applicability of any given system to any given scenario. We could of course do that for LI-COR systems, but again this is not the focus we want to give to our paper. We want to describe and quantify the problem, and let practitioners and companies figure out the details on a case-by-case basis.

It is almost certainly true that most $CO_2/H_2O$ measurement made with commercial solutions are not affected by this issue to a significant degree (although this is not always guaranteed and for this reason we can not state it). We pointed that out already in the first manuscript when we wrote "*Commercial solutions implementing sound engineering practices do exist for long-established EC of CO$_2$ and H$_2$O, which guarantee data synchronicity within specifications that meet EC requirements.* However, we don't think that this makes the problem any less relevant, especially because EC measurements of trace and ultra-trace gases are already challenging enough and - exactly because of the limited number of systems in use - they hardly receive the interest of commercial entities willing to implement industrial-grade *EC system* (as opposed to *gas analysis*) solutions for such species. In addition, we can/should expect interest in gas species other than $CO_2/H_2O$ to grow in time, and with that the number of "in-house" solutions. Already today, $CH_4$ and $N_2O$ are not so rarely measured: in ICOS, for example, they are mandatory for all Class 1 Stations (and the synchronization problem is indeed relevant to these stations).

*It would seem pertinent to put the errors reported here in the context of other sources of error, as suggested by the reviewers, to give the reader a sense of magnitude.*

We addressed this request by changing Figure 7 and its discussion accordingly (see also replies to Referee 2, Comment 6). We still think that it is more appropriate to talk about cut-off frequencies than about error amounts because cut-off frequencies are a property of the system, while flux attenuations also depend on the details of the deployment and of the turbulence at the site (as also evidenced in the first manuscript, where STE-induced errors were significantly different in different systems characterized by different cospectral content distribution) and for this reason we continued on this line improving, we hope, the clarity of figure and text.

**Replies to Referee 1**

*While the methods and conclusions seem sound, there is little practical information about how this work applies to real-world systems such as TK3, EasyFlux, EdiSol, HuskerFlux, or SmartFlux. This work would be of greater value if the authors could review some of these systems and comment on whether or not the issues they explore are present or absent in any of these packages.*

This is an important point that we also considered while preparing the manuscript. With this article, we wanted to present the problem of synchronization in fully-digital systems and characterize and quantify the corresponding errors. The aim was to reach out to developers of acquisition systems (industrial- or research-grade) as well as to EC practitioners in order to highlight the importance of this issue and its impact. In addition, synchronization issues are relevant not only to $CO_2$ and $H_2O$ but also to other gases. In fact, it can be presumed that they are even more relevant for gas species that to date received less industrial investment in terms of system integration, with the result that usually the data acquisition system must be designed and assembled by the researcher, given what is made available by the instruments' manufacturers. For this reason, instead of undertaking an analysis of the compliance for each acquisitions system today available or a quantification of its errors - which would have been extremely long, time consuming, incomplete and soon outdated - we focussed the manuscript on the description of the issue itself and propose a simple test for the evaluation of each given system. It can be noted, in fact, that even the acquisition system developed and commercialized by the company of two of the coauthors (SmartFlux™ by LI-COR Biosciences) was never cited in the original manuscript. In addition, since some of the solutions mentioned are commercial products, their evaluation could only be undertaken by collaborating with all involved parties. This is something certainly outside the scope of our work, but our hope is that this article will now enable such an analysis, providing a reference framework.

In the revised manuscript (Section 1.2), we provided reference to some of the existing solutions and attempted at putting them in context. As said, however, their evaluation remains outside the scope of the present work.

*1.) throughout the manuscript, "prospect" is used when "prospective" is appropriate*

Changed

*2.) on pg. 1, line 27, please define the term "zero-hold"*

That portion of the sentence was eliminated.

*3.) on pg. 5, line 11, change "AT clocks" to "AT cut crystals".... also throughout the manuscript, please don't confuse the term "clocks" (a system) with "crystals" (a component of a system).*

Corrected

*4.) on pg. 6, line10, what do the authors mean by the term "vector"? A vector is a quantity that has magnitude and direction. How does this apply to time?*

The sentence was reworded and the term "vector" eliminated.

*5.) on pg. 8, line 16, do the authors mean to use the term "filter" in this context? Does this imply that a mathematical operation was applied to the data in figure 5?*

We removed the word filter and used STE. We did characterize STEs as low-pass filters in the Materials and Methods section, so we were using the two words somewhat interchangeably in the original manuscript.

6.) on pg. 9, line 5, see #3 above

Corrected.

*The authors imply that all-digital data acquisition is a very recent development. This is not true. I've been aware of all-digital solutions for at least 15 years. One in particular (HuskerFlux from U. Nebraska or maybe Lawrence Berkeley Lab, I can't quite remember now) seems to have addressed a number of the issues identified here such as resynchronization of data streams.*

True, we changed the text removing references to the "recent development" and citing also HuskerFlux as example in the section 1.1. What is relatively recent is the use of a full digital acquisition system in the FLUXNET community but since we don't have quantitative information on this we remove the reference.

*The authors also imply that many of the synchronization problems outlined in the manuscript are absent from analog data acquisition systems, but this is not exactly true. Because of the "sample and hold" nature of A/D systems, many of these issues while present are masked.*

The Referee is correct. The measurement principle of the instruments covered by this paper is generally discrete-time (i.e. there is a defined "measurement interval" in each instrument). There is, in fact, an unavoidable zero order-hold in any analog output from such an instrument, which will be governed by the clock of the source instrument. When subsequently sampled by a "sample and hold" A/D system running on an independent clock, the sampling error that will occur will be of random nature (RTEs), and inversely proportional to the output rate of the analog signal (i.e. the higher the output rate, the smaller the resulting RTE). On the contrary, in traditional, mixed analog-digital system, development of STE is essentially impossible. We therefore added a brief note on the possible presence of random error in the introduction of the revised manuscript.

*The authors also suggest that Ethernet connectivity is also relatively new, but again, this has been available for a long time, especially in Campbell Scientific data loggers (via the NL-100 module).*
The reference to the Ethernet being relatively new has been removed

*One issue that the authors identify in serial data communications are the FIFO buffers used by many operating systems to ingest RS232 data. While these do exist and would create problems, well designed programs often get around this by lowering the size of these buffers and/or running independent program "threads" that handle individual character-received interrupts to pass the data along in near-real time.*

10 We do agree with the Referee that "well designed systems" can avoid the problems we present. In the Introduction, we also point out that "commercial solution exist […]". In the revised manuscript, we added and better clarified that well engineered solutions also exist and can be developed (by both commercial companies and non-commercial institutions), provided the appropriate engineering skills (electronics, computer science, digital signal processing, etc.) are available.

*The authors also state that STE timing issues are not detectable, but I must disagree. When testing several data acquisition packages, I found that the HuskerFlux package recorded the individual buffer size differences after a user chosen interval. This difference can be used to calculate the magnitude of the STE*
20 *over that interval. This should be relatively easy for any new software to do.*
We think the Referee is correct, but he is referring to the (potential) capabilities of the data logging system itself to monitor the traffic at the buffer and hence, at least in theory, determine the presence of STEs. But in that scenario, one may as well correct the STE, e.g. by re-interpolating the data, of which the actual frequency has been determined. What we refer to in the paper, instead, is the impossibility to establish
25 the presence of STEs in data files *"once acquired and stored in files"*, where wind and gas data have already been merged together (as stated in the original manuscript).

30 *Finally, in developing a method to check any particular system for timing errors, the authors suggest using a signal generator to inject a single waveform into the A/D input of both instruments while having one instrument also send the same signal from it's D/A outputs to the second instrument. While this will work in principle, it must be cautioned that this is only strictly true if the D/A task and the A/D task are both synchronized with the measurement task and the serial output task in both instrument firmwares.*
35 *Will this always be the case, or is this only true in some instruments such as the LiCor and Gill units*

*tested by the authors? If these tasks are only loosely synched or are running asynchronously, then some issues could be masked by the internal asynchronicity.*

It would seem that the Referee partially misunderstood the proposed test: "using the signal generator" and "using the analog signal of one instrument" are proposed as two alternative ways of reaching the same objective, they don't need to occur together. The important point in evaluating synchronization performance is to craft a means of producing a known signal, sampled independently by the two clocked systems. The resultant dataset, as captured by the proposed sampling system, can be directly evaluated, as the correlation should be perfect under these conditions. If we understood the Referee's comment, the concern is that imperfections upstreams may appear as deficiencies of the data logging system, while they are in fact issues of the instrumentation. We added a cautionary note on this in the revised manuscript. We also better explained that we proposed two different tests, one with the signal generator and the other with the analog output from the sensors.

*I would encourage the authors to consider these issues and perhaps explore some of the available all-digital solutions to perhaps assist users in choosing a system for their purposes.*

We have addressed this comment in previous parts of this documents.

**Replies to Referee 2**

*1. The opening of introductory section was too abbreviated. Recommend additional description/background of the EC method including additional references. The definition of synchronicity could be improved. For example, an EC system with a fixed, known lag would be considered asynchronous by this definition but not one that leads to flux error in the context of this manuscript.*

We extended the introduction with a richer description of the Eddy Covariance method and added references. We better clarified the definition of synchronicity referring explicitly to the "air parcel", which is really what matters. The case of a fixed lag is discussed in the time lags section and we better clarify that this case is solvable and so not further discussed in the paper.

*2. The manuscript has no explicit discussion of data triggering for digital data acquisition. This reviewer interpreted that all descriptions of digital data communications referred to streaming data. A brief discussion of triggering for data acquisition should be included in the Introduction, particularly as it relates to synchronizing data streams and timing errors.*

The Referee correctly interpreted that the paper is concerned with systems based on streamed data. As suggested, we added a brief discussion of systems based on data triggering and clarified that we concentrate on streaming-based systems (mainly in Section 1.2). The reason for this choice is that implementing a triggering-based system is much more complicated to the non-specialist and it is very often just not allowed by the instruments. For example, to the best of our knowledge most sonic anemometers and gas analysers (also beyond $CO_2$ and $H_2O$) do not support a triggering signal, while all instruments we are aware of do provide data in streaming mode. See also replies to the Editor.

*3. The manuscript (section 1.2) describes open digital communication protocols including serial and Ethernet (packet-based) but do not address SDM (Synchronous Device for Measurements) communications. SDM is a very commonly used data communication protocol for collecting EC measurements and eliminates many of the timing errors described in the manuscript through clock synchronization. The authors should include a discussion of this protocol and which timing errors are applicable.*

SDM does not "synchronize clocks", but rather implements a triggering strategy to avoid STEs (while still being subject to potential RTEs). It is therefore an instance of a system based on triggering signals, and we described it as such in the revised manuscript, including the caveat that instruments must be designed to work in triggering mode. For example, SDM protocol is a Campbell Scientific Inc. proprietary protocol and it is implemented in a small number of instruments (manufactured by only LI-COR Biosciences and Campbell Scientific) and, moving forward, it is likely to be adopted only on

Campbell's instrumentation. In general we tried to avoid explicit reference to specific commercial solutions (including SmartFlux™, released by LI-COR Biosciences), also because the list would be bound to be incomplete soon outdated. Therefore, in the revised manuscript we better differentiated the digital solutions separating streaming, triggering and polling modes, clarified that the issue mainly affects the streaming systems and added a short reference to SDM as one specific protocol for triggering-based systems. Most of these changes occurred in the Introduction, and specifically in Section 1.2.

*4. Considerable differences in flux errors (1 vs. 11%) were found between two sites given the same STE (180 μs/s). The explanation given was differences in the flux contribution in the frequency domain (cospectrum, see Figure 7) which is reasonable given the differences in observation height. In Fig 7, the cutoff frequency (transfer function) appears to differ between these two sites. However, in the text and as shown in Figure 6, the authors state that the transfer functions across sites were similar. Could this discrepancy be clarified?*

While we agree with the Referee and have been puzzled as well by the visually perceived difference between the two transfer functions implied in Fig. 7, we came to believe that the plot correctly shows the effect of the same transfer function on two cospectra with very different shapes. In the revised manuscript however, also following the comment of the Editor, we propose a new Figure that makes use of model cospectra, to provide more scenarios and related details about how and when STEs generate significant errors in the form of spectral losses. In that Figure, the same visual effect of Fig. 7 can be seen: compare, for example, plots (c) and (f).

*5. One the main points made in the manuscript is that timing errors cannot be corrected or detected a posteriori. Given that the authors frame timing errors in the context of a low pass filter, it seems reasonable (assuming of spectral similarity between w'T' and w'c') that timing errors would be accounted for and corrected by spectral correction methods that consider cospectra shape. Of course, such an approach could not differentiate between the source of signal loss (timing error, inlet tube attenuation, sensor separation, etc). The proposed approach assumes no timing error in the w'T' which is reasonable if calculated from a single SAT.*

We agree with the Referee that, in principle, an in-situ spectral correction method based on co-spectra would indeed correct timing errors as well. One caveat is that, in our opinion, spectral attenuations in the gas analyzer are better assessed using solely gas spectra rather than co-spectra (for the reasons put forward in Ibrom et al. 2007). It also appears to us that addressing the different sources of spectral losses individually allows for more control and fine-tuning of the correction. In the revised manuscript, we modified the discussion to better clarify this point and suggest that, at least in principle, a spectral

correction based on cospectra would indeed correct for STEs. We anyway also reiterated that one should always strive to avoid eliminable sources of bias errors, and STEs/RTEs fall in this category.

*6. The manuscript would be strengthened if the findings were placed in the context of other sources of EC errors and uncertainties, particularly for fluxes of gas species. For example, one could apply the timing-error transfer function to the gas cospectra in concert with transfer functions of other spectral loses to illustrate relative contributions.*

In the original manuscript (Pag. 8, Lines 22-27) we tried to put STE-induced errors in perspective by comparing the corresponding cut-off frequencies with those found in literature for modern $CO_2$/$H_2O$ EC systems. In general, we think that discussion of spectral losses of any given EC system should be limited to transfer functions and cut-off frequencies, because these are properties of the system itself, regardless of its deployment. Actual flux errors and relative contribution of different sources of spectral losses, as well as the other sources of uncertainty and error, depend critically on measurement height, turbulence regime and site characteristics. Nonetheless, following the suggestion of the Referee, in the revised manuscript we proposed a new Figure 7, for a more detailed and hopefully more intuitive, understanding of the errors potentially caused by STEs. . The figure makes use of model cospectra and shows how different STEs would affect the overall transfer function of a given EC system affected by other sources of spectral losses, in different deployment scenarios.

*Technical corrections:*

*Page 1, Line 30: Consider using specifications in place of specs*
Done.

*Page 2, Line 17: Use consistent spelling of analyzer throughout manuscript*
Done.

*Page 2, Line 27: Replace others with other*
Done.

*Page 4, Line 39: Missing reference to (Hewlett Packard 1997)*
Added.

*Page 5, Line 21: Acronyms (TCP) should be spelled out prior to use.*

Done.

*Page 7, Line 2: Incorrect symbol to denote range*
Changed with "-"

*Page 8, Line 9: 30 minute (not minutes) file*
Corrected.

*Page 8, Line 10: Incorrect symbol to denote range*
Changed with "-"

*Page 8, Line 24: Aand should be and.*
Corrected.

*Page 9, Lines 19-20: use consistent spelling for serial protocol used; previously authors used RS-232.*
Changed.

*Page 9, Line 23: Acronyms (OS) should be spelled out prior to use.*
Done.

*Page 10, Line 4: Native or naive?*
Naive

*Page 10, Line 12: Change to "thank".*
Done.

*Page 16: Clarify the figure and/or caption to denote that panel (a) illustrates transfer functions from a single site (IT-CA3).*
Caption changed.

*Page 20, Line 32: Authors cite the discussion version of Langford et al., 2015. Consider the final version (doi:10.5194/amt-8-4197-2015)?*
Updated.

*Page 21, Line 15: This reference (Smith 2002) was not cited in manuscript*

Eliminated.

**Other modifications from Short Interactive Comments**

5  *The authors should include more details about the fundamental principle of operation of the devices used in EC systems and more specifically the widely used NDIR analyzers … OMISSISS … They cannot provide a continuous analog signal, but rather discrete measurements. Consequently, the analog voltage outputs provided through the digital to analog converters (DAC) are discrete in time and magnitude.*

10  We agree with this comment and added a related short paragraph in the Introduction (section 1.4).

[revised manuscript text omitted]